# SECURING EQUAL SHARE: A PRINCIPLED APPROACH FOR LEARNING MULTIPLAYER SYMMETRIC GAMES

## ABSTRACT

This paper examines multiplayer symmetric constant-sum games with more than two players in a competitive setting, including examples like Mahjong, Poker, and various board and video games. In contrast to two-player zero-sum games, equilibria in multiplayer games are neither unique nor non-exploitable, failing to provide meaningful guarantees when competing against opponents who play different equilibria or non-equilibrium strategies. This gives rise to a series of long-lasting fundamental questions in multiplayer games regarding suitable objectives, solution concepts, and principled algorithms. This paper takes an initial step towards addressing these challenges by focusing on the natural objective of *equal share*—securing an expected payoff of $C/n$ in an $n$-player symmetric game with a total payoff of $C$. We rigorously identify the theoretical conditions under which achieving an equal share is tractable and design a series of efficient algorithms, inspired by no-regret learning, that *provably* attain approximate equal share across various settings. Furthermore, we provide complementary lower bounds that justify the sharpness of our theoretical results. Our experimental results highlight worst-case scenarios where meta-algorithms from prior state-of-the-art systems for multiplayer games fail to secure an equal share, while our algorithm succeeds, demonstrating the effectiveness of our approach.

## 1 INTRODUCTION

In recent years, AI systems have achieved remarkable success in multi-agent decision-making problems, particularly in a wide range of strategic games. These include, but are not limited to, Go (Silver et al., 2016), Mahjong (Li et al., 2020), Poker (Moravčík et al., 2017; Brown & Sandholm, 2018; 2019), Starcraft 2 (Vinyals et al., 2019), DOTA 2 (Berner et al., 2019), League of Legends (Ye et al., 2020), and Diplomacy (Gray et al., 2020; Bakhtin et al., 2022; , FAIR). Many of these games are two-player zero-sum games[1], where Nash equilibria always exist and can be computed in polynomial time. Nash equilibria in two-player zero-sum games are also non-exploitable—an agent employing a Nash equilibrium strategy will not lose even when facing an adversarial opponent who seeks to exploit the agent's weaknesses. Although such equilibrium strategies do not necessarily capitalize on opponents' weaknesses or guarantee large-margin victories, human players often adopt suboptimal strategies that deviate significantly from equilibria in complex games with large state spaces. Consequently, AI agents who adopt equilibrium strategies often outperform humans in practice for two-player zero-sum games.

In contrast, multiplayer games—defined here as those with *more than two players*—exhibit fundamentally different game structures compared to two-player zero-sum games. This distinction introduces several unique challenges. Firstly, Nash equilibria are believed to be no longer computable in polynomial time (Daskalakis et al., 2009; Chen & Deng, 2005). Moreover, there may exist multiple Nash equilibria with distinct values. Such non-uniqueness in equilibria raises a critical concern about the adoption of equilibrium strategies in multiplayer settings: if a learning agent adopts an equilibrium that is different from other players, collectively, they are not playing any single equilibrium, which undermines the equilibrium property that dissuades the agent from changing its strategy as long as others maintain theirs. Finally, in multiplayer games, equilibrium strategies

---

[1]Games such as DOTA and League of Legends, despite involving two teams, can be mostly considered similar to two-player zero-sum games in terms of their game structures and solutions.

are no longer non-exploitable and fail to provide meaningful guarantees when competing against opponents who are not playing equilibria. Although the introduction of alternative equilibrium notions such as (coarse) correlated equilibria alleviates computational hardness, issues of non-uniqueness and the lack of guarantees in the general settings remain. This leads to the first critical question:

**What is the *suitable solution concept* for learning in multiplayer games?**

Due to the presence of such fundamental challenge, even state-of-the-art expert-level or superhuman AI systems for popular multiplayer games, including Mahjong (Li et al., 2020) (4 players), Poker (Brown & Sandholm, 2019) (6 players), and Diplomacy (Bakhtin et al., 2022; , FAIR) (7 players), are designed with limited theoretical supports. These works focus on developing algorithmic frameworks capable of learning effective strategies that excel in ladders, online gaming platforms, or tournaments against opponents. Generally, most of these systems rely on a basic self-play framework, starting from scratch or from opponents' strategies acquired through behavior cloning, with or without regularization. While the success of these self-play algorithms is remarkable, their performance has been demonstrated primarily within the specific applications they were designed for, often coupled with human expertise and extensive engineering efforts. It remains unclear whether these algorithms are general-purpose solutions that can be readily applied to multiplayer games beyond Mahjong, Poker, or Diplomacy. This leads to the second important question:

**What is the *principled algorithm* that provable learns a rich class of multiplayer games?**

In this paper, we consider multiplayer symmetric constant-sum games, which are prevalent in games involving more than two players. Examples include previously discussed multiplayer games like Mahjong, Poker, and Diplomacy, as well as a variety of board games such as Avalon (Light et al., 2023), Mafia, and Catan.[2] *Symmetry brings fairness among players, providing a natural baseline where a learning agent should at least **secure an equal share** — achieving an expected payoff of $C/n$ in an $n$-player symmetric game with a total payoff of $C$. This paper takes an initial step toward addressing the two fundamental questions highlighted above by focusing on securing an equal share in multiplayer symmetric normal-form games[3]. The main contributions are summarized as follows:

1. Regarding the question of solution concepts, we first demonstrate the insufficiency of classical equilibrium concepts and general self-play frameworks (learning from scratch) in achieving an equal share in symmetric games. We then proceed to identify the structural conditions where equal share is achievable. In contrast to two-player zero-sum games, we prove that in order to achieve an equal share in multiplayer games: (1) all opponents need to deploy the same strategy; (2) all opponents must have limited adaptivity, and the learning agent has to model the opponents (See Section 4). We show that without either condition, an equal share can not be attained in the worst case. We prove that our identified conditions apply to practical multiplayer gaming platforms with a large player base. They are also tightly connected to the design of many prior modern multiplayer AI agents.

2. Regarding the question of principled algorithms, this paper illustrates how we can leverage existing tools from no-regret learning and no-dynamic-regret learning communities to achieve equal share with provable guarantees. Concretely, this paper considers several opponent settings: fixed, slowly adapting, and opponents that adapt at intermediate rates. For all cases, we design algorithms that approximately achieve equal share, with an error tolerance of $1/\text{poly}(T)$, where $T$ is the total number of games played. Additionally, we provide matching lower bounds, demonstrating that these guarantees cannot be significantly improved in the worst-case scenario.

3. We further complement our theory by experiments on two basic multiplayer symmetric games. Our experimental results illustrate that (1) the self-play meta-algorithms from prior state-of-the-art systems for multiplayer games can fail to secure an equal share even under favorable settings, while our principled algorithm always succeeds; (2) prior meta-algorithm has no clear advantage over our algorithm on exploitability in the worst case. This indicates that prior self-play algorithms are not general-purpose and highlights the effectiveness of our theoretical framework.

---

[2]All these examples are symmetric games up to randomization of the seating.

[3]Extensive-Form Games (EFGs) or Markov Games (MGs) can be viewed as special cases of normal-form games, where each action in normal-form games corresponds to a strategy in EFGs or MGs, although such representations may not always be efficient.

## 1.1 RELATED WORK

**AI gaming agents in practice**   Building superhuman AI has long been a goal in various games. A large body of works in this line focus on two-player or two-team zero-sum games like Chess (Campbell et al., 2002), Go (Silver et al., 2016), Heads-Up Texas Hold'em (Moravčík et al., 2017; Brown & Sandholm, 2018), Starcraft 2 (Vinyals et al., 2019), DOTA 2 (Berner et al., 2019) and League of Legends (Ye et al., 2020). Most of them are based on finding equilibria via self-play, fictitious play, league training, etc. There is comparatively much less amount of work on games with more than two players, whose game structures are fundamentally different from two-player zero-sum games. Several remarkable multiplayer successes include Poker (Brown & Sandholm, 2019), Mahjong (Li et al., 2020), Doudizhu (Zha et al., 2021) and Diplomacy (Bakhtin et al., 2022; , FAIR). Despite lacking a clearly formulated learning objective, these works typically design meta-algorithms, which include initially training the model using behavior cloning from opponents, then enhancing it through self-play, and finally applying adaptations based on the game's specific structure or human expertise. It remains elusive whether such a recipe is generally effective for a wide range of multiplayer games.

**Existing results for symmetric games**   Von Neumann & Morgenstern (1947) gave the first definition of symmetric games and used the three-player majority-vote example to showcase the stark difference between symmetric three-player zero-sum games and symmetric two-player zero-sum games. In his seminal paper that introduced Nash equilibrium, Nash proved that a symmetric finite multi-player game must have a symmetric Nash equilibrium (Nash, 1951). However, this existence result holds little significance from an individual standpoint, as there is no reason *a priori* to assume that other players are indeed playing according to this symmetric equilibrium. Papadimitriou & Roughgarden (2005) studied the computational complexity of finding the Nash equilibrium in symmetric multi-player games when the number of actions available is much smaller than the number of players and gave a polynomial-time algorithm for the problem. In this case, symmetry greatly reduced the computational complexity (as computing Nash in general is PPAD-hard). Daskalakis (2009) proposed *anonymous games*, a generalization of symmetric games.

**No-regret learning in games**   There is a rich literature on applying no-regret learning algorithms to learning equilibria in games. It is well-known that if all agents have no regret, the resulting empirical average would be an approximate Coarse Correlated Equilibrium (CCE) (Young, 2004), while if all agents have no swap-regret, the resulting empirical average would be an $\epsilon$-Correlated Equilibrium (CE) (Hart & Mas-Colell, 2000; Cesa-Bianchi & Lugosi, 2006). Later work continuing this line of research includes those with faster convergence rates (Syrgkanis et al., 2015; Chen & Peng, 2020; Daskalakis et al., 2021), last-iterate convergence guarantees (Daskalakis & Panageas, 2018; Wei et al., 2020), and extension to extensive-form games (Celli et al., 2020; Bai et al., 2022b;a; Song et al., 2022) and Markov games (Song et al., 2021; Jin et al., 2021).

## 2 PRELIMINARIES

**Notation.**   For any set $\mathcal{A}$, its cardinality is represented by $|\mathcal{A}|$, and $\Delta(\mathcal{A})$ denotes a probability distribution over $\mathcal{A}$. We employ $\mathcal{A}^{\otimes n}$ to denote the Cartesian product of $n$ instances of $\mathcal{A}$. Given a distribution $x$ over $\mathcal{A}$, $x^{\otimes n}$ represents the joint distribution of $n$ independent copies of $x$, forming a distribution over $\mathcal{A}^{\otimes n}$. For a function $f : \mathcal{A} \to \mathbb{R}$, we denote $\|f\|_{\infty} := \max_{a \in \mathcal{A}} |f(a)|$. We use $[n]$ to denote the set $\{1, \ldots, n\}$. In this paper, we use $C$ to denote universal constants, which may vary from line to line.

### 2.1 NORMAL-FORM GAMES AND EQUILIBRIUM

**Normal-form game**   An $n$-player normal-form game consists of a finite set of $n$ players, where each player has an action space $\mathcal{A}_i$ and a corresponding payoff function $U_i : \mathcal{A}_1 \times \cdots \times \mathcal{A}_n \to [-1, 1]$ with $U_i(a_1, \ldots, a_n)$ denotes the payoff received by the $i$-th player if $n$ players are taking joint actions $(a_1, \ldots, a_n)$. We define a game as constant-sum if there exists constant $C$ such that $\sum_{i=1}^{n} U_i(a_1, \ldots, a_n) = C$ for all joint actions. We further denote a game as zero-sum if it is a constant-sum game with a total payoff of $C = 0$. Normal-form games can represent a wide range of games as their special cases, including sequential games such as extensive-form or Markov games.

**Strategy**   A (mixed) strategy of a player is a probability distribution over the player's actions. For $i \in [n]$, we use $a_i \in \mathcal{A}$ and $x_i \in \Delta(\mathcal{A})$ to denote an action and a mixed strategy of the $i$-th player respectively. We use $a_{-i} \in \mathcal{A}^{\otimes n-1}$ and $x_{-i} \in \Delta(\mathcal{A}^{\otimes n-1})$ to denote the actions and the mixed strategies of the other players. We denote $U_i(x_i, x_{-i}) := \mathbb{E}_{a_i \sim x_i, a_{-i} \sim x_{-i}} [U_i(a_i, a_{-i})]$.

**Learning protocol**   We assume that the learner knows the game rule, and thus her own payoff function $U_1$. At every round $t$, all players take action simultaneously, and the learner only observes the opponents' noisy actions $(a_2^t, a_3^t, \ldots, a_n^t)$ that are sampled from their strategies.

**Best response**   Given a mixed strategy $x_{-i}$ of the other $n-1$ players, the *best response set* $\mathrm{BR}_i(x_{-i})$ of the $i$-th player is defined as $\mathrm{BR}_i(x_{-i}) := \arg\max_{a_i \in \mathcal{A}_i} U_i(a_i, x_{-i})$.

**Equilibrium   Nash Equilibrium (NE)** is the most commonly-used solution concept for games: a mixed strategy $x \in \Delta(\mathcal{A}_1 \times \cdots \times \mathcal{A}_n)$ of all players is said to be NE if $x$ is a product distribution [4], and no player could gain by deviating from her own strategy while holding all other players' strategies fixed. That is, for all $i \in [n]$ and $a_i' \in \mathcal{A}_i$, $\mathbb{E}_{a \sim x}[U_i(a_i, a_{-i})] \geq \mathbb{E}_{a \sim x}[U_i(a_i', a_{-i})]$.

There are also two equilibrium notions relaxing the notion of NE by no longer requiring $x$ to be a product distribution. It allows general joint distribution $x$ which describes correlated strategies among players. In particular, (1) $x$ is a **Correlated Equilibrium (CE)** if for all $i \in [n]$ and $a_i' \in \mathcal{A}_i$, $\mathbb{E}_{a \sim x}[U_i(a_i, a_{-i}) \mid a_i] \geq \mathbb{E}_{a \sim x}[U_i(a_i', a_{-i}) \mid a_i]$, and (2) $x$ is **Coarse Correlated Equilibrium (CCE)** if for all $i \in [n]$ and $a_i' \in \mathcal{A}_i$: $\mathbb{E}_{a \sim x}[U_i(a_i, a_{-i})] \geq \mathbb{E}_{a \sim x}[U_i(a_i', a_{-i})]$. The major difference between those two notions is in the cases when the agent deviates from her current strategy, whether she is still allowed to observe the randomness in drawing actions from the correlated strategy. The relationship among various equilibrium concepts is encapsulated by $\mathrm{NE} \subset \mathrm{CE} \subset \mathrm{CCE}$.

**Two-player zero-sum games**   It is well-known that in two-player zero-sum games, all Nash equilibria share the *unique* payoff value $0$. Furthermore, a Nash equilibrium is *non-exploitable* against any strategy that is not necessarily an equilibrium. In math, if $(\mu^\star, \nu^\star)$ is the Nash equilibrium, we have $\min_\nu U_1(\mu^\star, \nu) = \max_\mu \min_\nu U_1(\mu, \nu) = 0$.

**Multiplayer or general-sum games**   When the number of players is greater than two, or the games are no longer constant-sum games, Nash equilibria become PPAD-hard to compute (Daskalakis et al., 2009) while CEs and CCEs can be still computed in polynomial time. All of these three concepts admit multiple equilibria with distinct payoffs. Furthermore, they no longer own strong guarantees, such as non-exploitness, when competing against non-equilibrium players.

## 2.2   SYMMETRIC GAMES AND EQUAL SHARE

**Symmetric games**   For an $n$-player normal-form game with an action space $\{\mathcal{A}_i\}_{i=1}^n$ and a payoff $\{U_i\}_{i=1}^n$, we say the game is symmetric if (1) $\mathcal{A}_i = \mathcal{A}$, for all $i \in [n]$; (2) for any permutation $\sigma : [n] \to [n]$, we have $U_i(a_1, \cdots, a_n) = U_{\sigma^{-1}(i)}(a_{\sigma(1)}, \cdots, a_{\sigma(n)})$.

In short, the payoffs of a symmetric game for employing a specific action are determined solely by the actions used by others, agnostic of the identities of the players using them. Thus, the payoff function of the first player denoted as $U_1$, is sufficient to encapsulate the entire game.

Symmetric games are popular in practice as they bring fairness among players. Technically, all asymmetric can be converted to symmetric games by randomizing the roles of the players at the beginning of the game. Nevertheless, casting the games in the symmetric form gives a natural and minimal baseline — the learning agent should attain an equal share in the long run.

**Equal share**   We say an agent attains an equal share, if the agent's average payoff of the games is at least $C/n$ for a $n$-player symmetric constant-sum game with a total payoff of $C$.

It is not hard to see that shifting the total payoff of a game by any absolute constant will not alter the strategic aspects of the game. Therefore, without loss of generality, this paper sets a total payoff of $C = 0$ and focuses on achieving an equal share in **multiplayer symmetric zero-sum games**.

---

[4]The randomness in different players' strategies are independent.

---

**Algorithm 1** Self-play meta-algorithm

---

1: Initialize learner's mixed strategy $x_1$.
2: **for** $t = 1, \ldots, T$ **do**
3:     Sample action $a_i^t \sim x_t$ for all player $i \in [n]$.
4:     Update strategy $x_{t+1}$ using the gradient information $U_1(\cdot, a_{-1}^t)$.

---

## 3    INSUFFICIENCY OF EQUILIBRIA AND SELF-PLAY FOR EQUAL SHARE

In this section, we demonstrate that existing equilibria notions and the self-play from scratch algorithm are not sufficient to secure an equal share in multiplayer symmetric zero-sum games even under very basic settings. To illustrate this, we consider the following 3-player *majority vote* game:

**Example 1** (Three-player majority vote game). *Every player chooses either $0$ or $1$. If all players take the same action, then they receive a payoff of 0. Otherwise, being in the majority yields a positive payoff of $1/2$, while being in the minority results in a negative payoff of $-1$.*

**Insufficiency of equilibria**     In this setup, both pure strategies $(0, 0, 0)$ and $(1, 1, 1)$ constitute NE. However, the existence of multiple NEs creates a predicament for the learning agent. It must choose which equilibrium to follow, yet there is always the risk that the two opponents are both playing the other NE, leading to a negative payoff for the learner. In other words, adhering to a single NE does not reliably ensure an equal share when multiple equilibria exist. Since NE $\subset$ CE $\subset$ CCE, we know the same limitation also holds for CE and CCE.

**Insufficiency of self-play from scratch**     Self-play is a training method in which the learning agent improves its performance by repeatedly playing against copies of itself without human supervision. See pseudo-code in Algorithm 1. The learner maintains its own strategy $\{x_t\}_{t=1}^T$. At the $t^{\text{th}}$ iteration, the learner first pretends that all opponents are employing its current strategy $x_t$, and samples actions from them. Then the learner updates its own strategy to $x_{t+1}$ using the gradient information from the gameplay. The updates can be made using any optimizer such as gradient descent or Hedge.

Here, we argue that self-play from scratch (the algorithm adopted in Brown & Sandholm (2019)) again fails to secure an equal share in the same three-player majority vote game: Consider two opponents play the same fixed strategy that is one of the NEs. In this case, the learner has no choice but to play the exact same NE as the opponents to secure an equal share. That is, if the learner's algorithm is agnostic to the strategies of the opponents, it is doomed to fail. We note that while recent systems (Li et al., 2020; Jacob et al., 2022) combine self-play with behavior cloning which is no longer agnostic to opponents' strategies, our experiment shows that their meta-algorithms remain insufficient to secure an equal share in the worst-case scenario (See Section 6).

## 4    SUFFICIENT CONDITIONS FOR SECURING EQUAL SHARE

In this section, we identify the structural conditions of the games where equal share is achievable. We will show that the following two conditions are needed to achieve equal share:

**Condition 1.** All opponents need to deploy the same strategy, i.e., $x_2 = \ldots = x_n$;

**Condition 2.** All opponents must have limited adaptivity and the player has to model the opponents, i.e., $\{x_j\}_{j=2}^n$ can not adversarially change across different rounds of the game.

We justify these two conditions by proving that without either condition, an equal share can not be attained in multiplayer symmetric games in the worst case. We remark that both conditions restrict the strategies of opponents rather than the type of games.

**Non-adaptive opponents that deploy different strategies**     We start by considering the case where Condition 2 holds but Condition 1 does not hold.

**Proposition 4.1.** *There exist symmetric zero-sum games with opponents using fixed but differing strategies, such that* no *learner's strategy secures an equal share. In math,*

$$\max_{x_1} \min_{x_2, \cdots, x_n} U_1(x_1, \cdots, x_n) \leq \min_{x_2, \cdots, x_n} \max_{x_1} U_1(x_1, \cdots, x_n) \leq 0 \qquad (1)$$

*where both inequalities can be made strict in certain games.*

Here, we can further strengthen the proposition to require opponents to deploy strategies without "collusion". That is, the hard instance holds even when the strategies employed by opponents are statistically independent without any shared randomness. Proposition 4.1 highlights the challenge when opponents are free to adopt different strategies.

**Adaptive opponents that deploy identical strategy**    We next examine the case where Condition 1 holds but Condition 2 does not hold.

**Proposition 4.2.** *There exist symmetric zero-sum games such that* no *learner's strategy secures an equal share against adversarial opponents, even under the constraint that they adhere to identical strategy at each round. In math,*

$$\max_{x_1} \min_x U_1(x_1, x^{\otimes n-1}) \leq \min_x \max_{x_1} U_1(x_1, x^{\otimes n-1}) = 0, \tag{2}$$

*where the inequality can be made strict in certain games.*

Proposition 4.2 implies a property that makes multiplayer games significantly different from two-player zero-sum games: even under the favorable scenario of all opponents employing identical strategies, one can no longer find a fixed "non-exploitable" strategy agnostic to the strategies of opponents. All strategies are exploitable. Opponent modeling is necessary to secure an equal share.

**Solution concepts beyond equilibrium**    Combining Proposition 4.1 and Proposition 4.2, we observe that both conditions mentioned at the beginning of Section 4 are needed to make equal share achievable. In math, we conclude from Eq.(1) and Eq.(2) that, out of the four related minimax concepts, only $\min_x \max_{x_1} U_1(x_1, x^{\otimes n-1}) = 0$ across all multiplayer symmetric zero-sum games, which guarantees an equal share. This remaining minimax concept precisely corresponds to the two identified conditions where opponents employ identical strategies, and the learner must be adaptive to the opponents. Therefore, we will use $\min_x \max_{x_1} U_1(x_1, x^{\otimes n-1})$ as our **target solution concept** for this paper to achieve equal share. We remark that *this solution concept does not necessarily correspond to any equilibrium in most multiplayer games*. We conclude with this solution concept from a principled manner with equal share as our primary objective. For conciseness, from now on, we will also refer to the common strategy employed by all opponents as the **meta-strategy**.

## 4.1    CONNECTIONS BETWEEN IDENTIFIED CONDITIONS AND PRACTICE

While the two identified conditions may seem restrictive, here we argue that they in fact apply to practical multiplayer gaming platforms with a large player base. Condition 1 is further implicitly adopted by most prior state-of-the-art AI agents for multiplayer games.

**Connection to multiplayer games with a large player base**    We argue that both identified conditions are well-justified in modern multiplayer gaming platforms with a large player base. Imagine a casino hosting $N$ players who randomly join poker tables or an online Mahjong match-making platform with $N$ users. Let $\{x_i\}_{i=1}^N$ be the strategy set for these $N$ players. We can then define the *population meta-strategy* as $\bar{x} = (1/N) \sum_{i=1}^N x_i$. The following proposition claims that, for $n$-player symmetric zero-sum games, in terms of the expected payoff, playing against $n-1$ random players is almost the same as playing against $n-1$ players who all adopt the same population meta-strategy $\bar{x}$, as long as $N \gg (n-2)^2$.

**Proposition 4.3.** *Let $\mathbb{E}_{x_{-1}}$ be the expectation over the randomness on sampling $n-1$ strategies uniformly from the set $\{x_i\}_{i=1}^N$ without replacement. Then for any strategy $z \in \Delta(\mathcal{A})$, we have $|\mathbb{E}_{x_{-1}}[U_1(z, x_{-1})] - U_1(z, \bar{x}^{\otimes n-1})| \leq 2(n-2)^2/N$.*

Furthermore, it is often safe to assume that the population meta-strategy $\bar{x}$ — the average strategy of all players within the player pool — will not quickly adapt to one particular player's strategy.

**Connection to practical AI systems**    We remark that a majority of practical AI systems for multiplayer games (Li et al., 2020; Brown & Sandholm, 2019; Bakhtin et al., 2022; , FAIR) leverage self-play meta-algorithms (Algorithm 1), which equate the strategies of all opponents with those of the learner. This implicitly assumes all opponents employ an identical strategy at every round.

# 5 PROVABLY EFFICIENT ALGORITHMS

In this section, we explore efficient algorithms that provably secure an equal share under the two conditions identified in Section 4. Particularly, we consider several opponent settings with various adaptivity: fixed, slowly adapting, and opponents that adapt at intermediate rates.

We use the following notations throughout this section: at round $t$, let $x^t$ denote the learner's strategy, and $y^t$ the meta-strategy employed by all opponents. We denote $u^t(\cdot)$ as the expected payoff function of the learner at round $t$, where $u^t(\cdot) := U_1(\cdot, (y^t)^{\otimes n-1})$. Then the average payoff of the learner is:

$$u_{\text{avg}}(T) := (1/T) \sum_{t=1}^{T} u^t(x^t).$$

## 5.1 FIXED OPPONENTS

We begin by exploring the simple stationary scenario, where the meta-strategy used by the opponents remains constant over time, denoted as $y^t = y$ for all $t \in [T]$.

Notably, in this particular scenario, the payoff function $u^t(\cdot)$ remains constant over time. Additionally, by symmetry, it is not hard to observe that at least one action will consistently yield an expected payoff of 0 in all rounds. This implies $\max_{a \in \mathcal{A}} \sum_{t=1}^{T} u^t(a) \geq 0$, which makes the no-regret learning tool well-poised to achieve equal share in this setting. Standard *(static) regret*, defined as follows, compares the learner's total payoff to the total payoff achieved by the best action in hindsight:

$$\text{Reg}(T) := \max_{a \in \mathcal{A}} \sum_{t=1}^{T} u^t(a) - \sum_{t=1}^{T} u^t(x^t),$$

An algorithm has *no-regret* if $\text{Reg}(T) \leq o(T)$ for all large $T$. We deploy a standard no-regret learning algorithm — Hedge (Freund & Schapire, 1997), which provides the following guarantees:

**Theorem 5.1** (Stationary opponents). *Let $\{x^t\}_{t=1}^{T}$ be the strategy sequence implemented by the Hedge algorithm against stationary opponents. Then, with probability at least $1 - \delta$, we have*

$$u_{avg}(T) \geq u^\star - C\sqrt{\log(A/\delta)/T},$$

*for some absolute constant $C$, where $u^\star := \max_{a \in \mathcal{A}} U_1(a, y^{\otimes n-1}) \geq 0$.*

The probability is taken over the random actions by opponents sampled from the meta-strategy[5]. Theorem 5.1 claims that with stationary opponents, the Hedge algorithm approximately achieves an equal share up to a $\tilde{O}(1/\sqrt{T})$ error, which demonstrates its effectiveness in the long run.

## 5.2 ADAPTIVE OPPONENTS

In practical scenarios, encountering a fixed opponent strategy is relatively uncommon. More often, opponents adapt and modify their strategies over time, responding to the game's dynamics and the actions of other players. Thus, in this section, we shift our focus to the non-stationary scenario, where the meta-strategy $y^t$ adopted by the opponents *varies* over time.

According to Proposition 4.2, it is clear that attaining an equal share is impossible if opponents can change their meta-strategy $y^t$ arbitrarily fast across different rounds. Thus we introduce a constraint on the adaptive power of the opponents by positing a variation budget $V_T$, which bounds the total variation of the payoff function across the time horizon. Specifically, we assume the payoff function belongs to $\mathcal{U}$, which is defined as

$$\mathcal{U} := \left\{ \{u^t\}_{t=1}^{T} \; \middle| \; \sum_{t=1}^{T-1} \left\| u^{t+1} - u^t \right\|_\infty \leq V_T \right\}. \tag{3}$$

Furthermore, we denote $\mathcal{G}(n, A, V_T)$ as the set of tuples, which consists of a $n$-player symmetric zero-sum game with $A$ actions and a corresponding meta-strategy sequence $\{y^t\}_{t=1}^{T}$, such that the payoff function $\{u^t\}_{t=1}^{T} \in \mathcal{U}$. This constraint effectively moderates the adaptivity of the opponents compared to a fully adversarial setup.

---

[5]Recall that our learning protocol in Section 2 assumes the learner only observes the noisy actions of the other players at each round.

**Slowly adapting opponents**   In non-stationary environments, the total payoff achieved by the best action in hindsight $\max_{a \in \mathcal{A}} \sum_{t=1}^{T} u^t(a)$ is no longer non-negative. Therefore, minimizing standard regret in this setting is no longer effective in securing an equal share. This motivates us to turn our attention to a stronger notion of regret — *dynamic regret*, defined as:

$$\text{D-Reg}(T) := \sum_{t=1}^{T} \max_{a \in \mathcal{A}} u^t(a) - \sum_{t=1}^{T} u^t(x^t).$$

This measures a strategy's performance against the best action at each time step (dynamic oracle), providing a more relevant benchmark in changing environments.

In the setting of symmetric games, the dynamic oracle is always assured to secure an equal share, i.e., $\sum_{t=1}^{T} \max_{a \in \mathcal{A}} u^t(a) \geq 0$. Thus, any algorithm achieving no-dynamic-regret is guaranteed to achieve an equal share up to a small error. To this ends, we adapt a no-dynamic-regret algorithm— Strongly Adaptive Online Learner with Hedge $\mathcal{H}$ as a black box algorithm ($\text{SAOL}^{\mathcal{H}}$) (See Appendix C.1.2), as proposed by Daniely et al. (2015) to our setting and achieve following guarantees:

**Theorem 5.2.** *Suppose that $n \geq 3$, $A \geq 2$, and $V_T \in [1, T]$, then for any game and any meta-strategy sequence in $\mathcal{G}(n, A, V_T)$, with probability at least $1 - \delta$, $\text{SAOL}^{\mathcal{H}}$ satisfies*

$$u_{avg}(T) \geq u^{\dagger} - C V_T^{1/3} T^{-1/3} \left( \sqrt{\log(A/\delta)} + \log T \right)$$

*for some absolute constant $C$, where $u^{\dagger} := (1/T) \sum_{t=1}^{T} \max_{a \in \mathcal{A}} u^t(a) \geq 0$.*

Theorem 5.2 implies that $\text{SAOL}^{\mathcal{H}}$ achieves a non-negative average payoff, up to an error term that scales with $\tilde{O}(V_T^{1/3} T^{-1/3})$. Therefore, if $V_T$ is sublinear in $T$, $\text{SAOL}^{\mathcal{H}}$ is capable of approximately achieving equal share over an extended duration.

**Opponents that adapt at intermediate rates**   Interestingly, there is an intermediate regime where opponents' strategies $\{y^t\}_{t=1}^{T}$ are changing neither too fast nor too slow where the favorable algorithm for the learner might be simply behavior cloning—simply mimic opponents' strategies.

Formally, we define the behavior cloning algorithm for the learner by making her action in $t$-th round the same as the action taken by the 2nd player in $(t-1)$-th round (See Algorithm 2). Behavior cloning achieves the following:

**Theorem 5.3.** *Suppose that $n \geq 3$, $A \geq 2$, and $V_T \in [1, T]$, for any game and any meta-strategy sequence in $\mathcal{G}(n, A, V_T)$, behavior cloning guarantees that*

$$\mathbb{E}[u_{avg}(T)] \geq -(V_T + 1)/T.$$

We remark that while the error term $O(V_T/T)$ in Theorem 5.3 is always smaller than the error term $\tilde{O}((V_T/T)^{1/3})$ in Theorem 5.2, the latter is comparing to the baseline of dynamic oracle $u^{\dagger}$, which has a greater value than the baseline in behavior cloning — 0. It is not hard to see that in the intermediate regime $\tilde{\Theta}(u^{\dagger}) \leq V_T/T \leq \Theta(1)$, the meta-strategy is changing too fast so that it is better for the learner to simply copy the meta-strategy instead of running sophisticated no-dynamic-regret algorithm to learn the game and to counter the meta-strategy by herself.

**Matching lower bounds**   Finally, we also complement our upper bounds by matching lower bounds showing that $\text{SAOL}^{\mathcal{H}}$ and behavior cloning are already the near-optimal algorithms in terms of error rates when compared with the corresponding baselines — the dynamic oracle and zero payoff respectively. The techniques used here are based on adapting existing hard instances for a more general setup to the symmetric zero-sum game setting. Please see more discussion in Appendix C.2.

**Theorem 5.4.** *There exists some absolute constant $C > 0$ such that for any $n \geq 3$, $A \geq 2$, and $V_T \in [1, T]$, and any learning algorithm, there exists a game and a meta-strategy sequence in $\mathcal{G}(n, A, V_T)$, such that $\mathbb{E}[u_{avg}(T)] \leq u^{\dagger} - C V_T^{1/3} T^{-1/3}$.*

**Theorem 5.5.** *There exists some absolute constant $C > 0$ such that for any $n \geq 3$, $A \geq 2$, and $V_T \in [1, T]$, and any learning algorithm, there exists a game and a meta-strategy sequence in $\mathcal{G}(n, A, V_T)$ such that $\mathbb{E}[u_{avg}(T)] \leq -C V_T/T$.*

The expectation in both theorems are taking over the random actions by the opponents as well as the possible intrinsic randomness in a stochastic algorithm. Theorem 5.4 and 5.5 match Theorem 5.2 and 5.3 respectively, up to additional logarithmic factors.

## 6 EXPERIMENTS

In this section, we focus on the scenario where one learning agent competes against $n-1$ opponents who play the identical meta-strategy. For simplicity, we restrict ourselves to the setting of fixed opponents. We aim to answer: **(Q1)** Can existing algorithmic frameworks in previous superhuman AI systems consistently secure an equal share under this favorable setting? If not, what are the failure cases? **(Q2)** Are these trained agents exploitable by adversarial opponents? We design the following two games to compare our algorithm with prior self-play-based algorithms.

**Majority Vote (MV).** We first consider the standard 3-player majority vote game (Example 1). It is not hard to see that $[1, 0]$, $[0, 1]$, and $[1/2, 1/2]$ are all NEs, where $[p, 1 - p]$ denotes the mixture strategy that takes the first action with probability $p$ and the second action with probability $1 - p$. We fix the opponents' meta-strategy $y_{\mathrm{meta}} = [0.49, 0.51]$ for all rounds.

**Switch Dominance Game (SDG).** In each round, players simultaneously choose an action from set $\{A, B, C\}$. Let $n$ be the total number of players and $n_A$ be the number of agents choosing action $A$, We define the game rule as:

$$\begin{cases} B \succ A \succ C & \text{if } n_A > 0.2n, \\ C \succ B \succ A & \text{otherwise}, \end{cases}$$

where the rule $i \succ j \succ k$ intuitively means that action $i$ dominates both $j$ and $k$, and action $j$ dominates $k$. SDG is designed so that $C$ is a *dominated* action when there is a reasonable number of players taking action $A$, but a *dominating* action otherwise. Concretely, for $i \succ j \succ k$, we assign the following payoff $(r_i, r_j, r_k)$ to players taking actions $(i, j, k)$ respectively, where:

$$r_i = \mathbb{I}[n_j + n_k > 0], \quad r_j = \mathbb{I}[n_k > 0] - \mathbb{I}[n_j + n_k > 0] \cdot n_i/(n_j + n_k),$$
$$r_k = -\mathbb{I}[n_j + n_k > 0] \cdot n_i/(n_j + n_k) - \mathbb{I}[n_k > 0] \cdot n_j/n_k.$$

This payoff design guarantees that SDG is a symmetric zero-sum game. Throughout our experiments, we choose $n = 30$ and pick the fixed meta-strategy of the opponents $y_{\mathrm{meta}} = [0.399, 0.6, 0.001]$ (in the order of action $A, B, C$) for all rounds. Note that while this game has an NE strategy $[0, 0, 1]$, its utility is negative against our chosen meta-strategy $y_{\mathrm{meta}}$.

### 6.1 LEARNING ALGORITHMS

To better focus on the key game-theoretic property of the algorithms, we idealize the process of imitation learning by assuming that the learning agent has already learned (i.e., has direct access to) the meta-strategy $y_{\mathrm{meta}}$ by the opponents. In this setting, according to Theorem 5.1, our theoretical framework suggests to directly run the *Hedge* algorithm (Hedge) against opponents who play the meta-strategy. We compare our algorithm against three meta-algorithms adopted by prior state-of-the-art AI systems in practice: (1) *self-play from scratch* (SP_scratch) (Brown & Sandholm, 2019); (2) *self-play initialized from behavior cloning* (SP_BC) (Li et al., 2020), and (3) *self-play initialized from behavior cloning with regularization towards the meta-strategy* (SP_BC_reg) (Jacob et al., 2022). While these AI systems further implement multi-step lookahead with a few additional techniques, many of them only apply to sequential games, not the basic normal-form games. Here, we focus on the comparison of the high-level game-theoretic meta-algorithms.

**Algorithm details.** We also use the Hedge algorithm as the optimizer for the self-play algorithm to update its strategy. We choose the learning rate for the Hedge algorithm based on theoretically optimal value and choose the regularization parameter according to Jacob et al. (2022). We refer readers to Appendix D for more details.

### 6.2 RESULTS

To answer **Q1** and **Q2**, we evaluate the utility of the learned strategy $\hat{x}$ against the pre-specified meta-strategy $y_{\mathrm{meta}}$, i.e., $U_1(\hat{x}, y_{\mathrm{meta}}^{\otimes(n-1)})$, as well as the exploitability of the learner $\hat{x}$, i.e., $\min_y U_1(\hat{x}, y^{\otimes(n-1)})$. To measure the utility, we evaluate the payoff of the agent's converged strategy by Monte Carlo methods with $3 \times 10^5$ games and report the mean and standard deviation of 10 runs. As for the exploitability, pick the best exploiter strategy within 100 runs, and report the payoff of the learner against that exploiter.

| Strategy | SP_scratch | SP_BC | SP_BC_$10^{-5}$ | SP_BC_$10^{-4}$ | SP_BC_$10^{-3}$ | SP_BC_$10^{-2}$ | **Hedge** |
|---|---|---|---|---|---|---|---|
| $[1, 0]$ | 52% | 48% | 50% | 46% | 42% | 42% | 0% |
| $[0, 1]$ | 48% | 52% | 50% | 54% | 58% | 58% | 100% |

Table 1: The distribution of mixed strategies to which different self-play algorithms converge in MV. We use SP_BC_$\lambda$ as the short name of SP_BC_reg with regularization coefficients $\lambda$.

| MV | SP_scratch / SP_BC / SP_BC_reg | **Hedge** |
|---|---|---|
| Utility ($\times 10^{-2}$) | $-1.00 \pm 0.09$ | $\mathbf{1.03} \pm 0.10$ |
| Exploitability | $-1.00 \pm 0.00$ | $-1.00 \pm 0.00$ |

| SDG | SP_scratch / SP_BC / SP_BC_reg | **Hedge** |
|---|---|---|
| Utility | $-12.67 \pm 0.01$ | $\mathbf{1.00} \pm 0.00$ |
| Exploitability | $-29.00 \pm 0.00$ | $-29.00 \pm 0.00$ |

Table 2: The utility and exploitability of each algorithm. Particularly, for MV, as self-play algorithms converge to two different solutions with roughly equal probability, we evaluate the utility of the worse converged solution of the two to reflect the performance in the worst case.

**Convergence analysis.** We first check the convergence for each algorithm in both games:

**MV**: We report the limiting solution each algorithm converges to within 100 runs, as summarized in Table 1. Our algorithm (Hedge) consistently converges to the good strategy $[0, 1]$. All self-play variants have significant probability converge to the bad strategy $[1, 0]$, which has a negative utility against the chosen meta-strategy $y_{\text{meta}} = [0.49, 0.51]$.

**SDG**: We report that while our Hedge algorithm converges to the strategy $[0, 1, 0]$, all self-play variants consistently converge to the strategy $[0, 0, 1]$.

**Utility and Exploitability.** We summarize the results in Table 2, which show that even in these two simple symmetric zero-sum games, none of the self-play algorithms can consistently secure a non-negative payoff, i.e., an equal share, in the worst case. This undesirable behavior persists even without opponents making any adaptations! Moreover, based on these two games, we further conclude two potential failure modes of self-play algorithms: (1) For games with multiple NEs, such as MV, self-play methods may converge to different NEs based on different initialization. When the opponents' meta-strategy (i.e., the initial strategy for SP_BC) lies close to the boundary of the convergence basins of two different NEs, self-play algorithms will have a non-zero probability of converging to both of them due to the statistical randomness in the game. It is likely one of the two NEs is undesirable against the meta-strategy $y_{\text{meta}}$. (2) For games with a single NE, self-play algorithms are still very likely to be attracted to this equilibrium. A carefully designed game structure can result in this NE yielding a negative utility against the chosen meta-strategy, and hence jailbreak all self-play variants. The aforementioned failure modes highlight a significant limitation of self-play variants when being applied to diverse and complex multiplayer games. In contrast, the principled algorithm according to our theory consistently beats the meta-strategy of the opponents, receives a much higher payoff, and secures an equal share. Regarding exploitability, all learned strategies can be easily exploited by adversarial opponents.

# 7 CONCLUSION

Unlike in the two-player zero-sum games, standard equilibria are no longer always the suitable solution concept for multiplayer games. They are non-unique and lack meaningful guarantees in general scenarios. This paper establishes a new theoretical framework that provides the solution concepts and the principled algorithms for multiplayer games from the unique angle of achieving equal share. We hope our results serve as the first step toward further research on principled methodologies and algorithms for multiplayer games.

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

# A EXTENDED PRELIMINARIES

## A.1 NO-REGRET LEARNING

No-regret learning is a commonly adopted strategy in game theory to find equilibrium solutions. We consider a $T$-step learning procedure, where for each round $t \in [T]$: (1) the agent picks a mixed strategy $\mu^t$ over $\mathcal{A}$, (2) the environment picks an adversarial loss $\ell_t \in [0,1]^{|\mathcal{A}|}$. The expected utility for $t$-th round is defined as $-\langle \mu^t, \ell_t \rangle$. To measure the performance of a particular algorithm, a common approach is to consider regret, where the algorithm's performance is compared against the single best action in hindsight. Specifically, for policy sequence $(\mu^1, \ldots, \mu^T)$ taken by an algorithm, the static regret is given, by

$$\text{Reg}(T) = \sum_{t=1}^{T} \langle \mu^t, \ell_t \rangle - \min_{a \in \mathcal{A}} \sum_{t=1}^{T} \ell_t(a).$$

We say that the algorithm is a no-regret algorithm if $\text{Reg}(T) = o(T)$. One of such no-regret learning algorithms is **Hedge algorithm**, which performs the following exponential weight updates:

$$\mu^{t+1}(a) \propto \mu^t(a) e^{-\eta_t \ell_t(a)}, \quad \text{for} \quad \forall a \in \mathcal{A}.$$

where $\eta_t$ is the learning rate. See Algorithm 5 for the Hedge algorithm as applied to our problem setup.

## A.2 3-PLAYER MAJORITY AND MINORITY GAME

In this section, we give a formal definition of the 3-player majority and minority game.

We define the 3-player *majority game* as a symmetric zero-sum game with action space $\mathcal{A} := \{0, 1\}$ and the payoff function given by:

$$U_1(0,0,0) = U_1(1,1,1) = 0$$
$$U_1(0,1,0) = U_1(0,0,1) = U_1(1,1,0) = U_1(1,0,1) = 1/2$$
$$U_1(0,1,1) = U_1(1,0,0) = -1.$$

In other words, players receive a positive payoff if they are part of the majority and a negative payoff if they are in the minority. Correspondingly, we define the 3-player *minority game* as a symmetric zero-sum game with action space $\mathcal{A} := \{0, 1\}$ and the payoff function given by:

$$U_1(0,0,0) = U_1(1,1,1) = 0$$
$$U_1(0,1,0) = U_1(0,0,1) = U_1(1,1,0) = U_1(1,0,1) = -1/2$$
$$U_1(0,1,1) = U_1(1,0,0) = 1.$$

In other words, players receive a positive payoff if they are part of the minority and a negative payoff if they are in the majority.

# B PROOFS FOR SECTION 4

In the sequel, we will prove Proposition 4.1 in Section B.1, Proposition 4.2 in Section B.2 and Propostion 4.3 in Section B.3.

## B.1 PROOF OF PROPOSITION 4.1

In this section, we will prove (1), where both inequalities can be made strict in certain games.

*Proof of Proposition 4.1.* For the first inequality, note that for any $(x_1, \ldots, x_n)$:

$$U_1(x_1, \cdots, x_n) \le \max_{x_1} U_1(x_1, \cdots, x_n),$$

which implies

$$\min_{x_2, \cdots, x_n} U_1(x_1, \cdots, x_n) \le \min_{x_2, \cdots, x_n} \max_{x_1} U_1(x_1, \cdots, x_n).$$

By further taking maximum over $x_1 \in \Delta(\mathcal{A})$, we prove that

$$\max_{x_1} \min_{x_2, \cdots, x_n} U_1(x_1, \cdots, x_n) \leq \min_{x_2, \cdots, x_n} \max_{x_1} U_1(x_1, \cdots, x_n).$$

To show the first inequality can be strict, we consider the 3-player majority vote. Suppose 3 players adopt the mixed strategies $(\alpha_1, 1 - \alpha_1)$, $(\alpha_2, 1 - \alpha_2)$ and $(\alpha_3, 1 - \alpha_3)$, respectively. It then holds that

$$U_1(x_1, x_2, x_3) = U_1(\alpha_1, \alpha_2, \alpha_3)$$

$$= \alpha_1 \left( -(1 - \alpha_2)(1 - \alpha_3) + \frac{1}{2}\alpha_2(1 - \alpha_3) + \frac{1}{2}\alpha_3(1 - \alpha_2) \right)$$

$$+ (1 - \alpha_1) \left( -\alpha_2\alpha_3 + \frac{1}{2}\alpha_2(1 - \alpha_3) + \frac{1}{2}\alpha_3(1 - \alpha_2) \right).$$

By choosing $\alpha_2 = \alpha_3 = 0$ when $\alpha_1 > 1/2$ and $\alpha_2 = \alpha_3 = 1$ when $\alpha_1 \leq 1/2$, it can be seen that

$$\max_{\alpha_1} \min_{\alpha_2, \alpha_3} U_1(\alpha_1, \alpha_2, \alpha_3) \leq \max_{\alpha_1} \min\{-\alpha_1, -(1 - \alpha_1)\} = -\frac{1}{2}.$$

Note that

$$\min_{\alpha_2, \alpha_3} \max_{\alpha_1} U_1(\alpha_1, \alpha_2, \alpha_3)$$

$$= \frac{1}{2} \min_{\alpha_2, \alpha_3} \max\{-2(1 - \alpha_2)(1 - \alpha_3) + \alpha_2(1 - \alpha_3) + \alpha_3(1 - \alpha_2),$$

$$- 2\alpha_2\alpha_3 + \alpha_2(1 - \alpha_3) + \alpha_3(1 - \alpha_2)\}$$

$$= \frac{1}{2} \min_{\alpha_2, \alpha_3} \max\{3(\alpha_2 + \alpha_3) - 4\alpha_2\alpha_3 - 2, \alpha_2 + \alpha_3 - 4\alpha_2\alpha_3\}$$

$$= 0.$$

Thus, we show that $\max_{\alpha_1} \min_{\alpha_2, \alpha_3} U_1(\alpha_1, \alpha_2, \alpha_3) < \min_{\alpha_2, \alpha_3} \max_{\alpha_1} U_1(\alpha_1, \alpha_2, \alpha_3)$, which implies the first inequality can be strict.

For the second inequality, due to a restriction on the minimization constraints, it is straightforward that

$$\min_{x_2, \cdots, x_n} \max_{x_1} U_1(x_1, \cdots, x_n) \leq \min_x \max_{x_1} U_1(x_1, x^{\otimes n-1}).$$

In the sequel, we prove $\min_x \max_{x_1} U_1(x_1, x^{\otimes n-1}) = 0$ via contradiction. Note that by choosing $x_1 = x$, we can show that

$$\min_x \max_{x_1} U_1(x_1, x^{\otimes n-1}) \geq 0.$$

Suppose for some game inequality holds, then by definition

$$\forall x \in \Delta(\mathcal{A}), \exists x' \in \Delta(\mathcal{A}), s.t. U_1(x', x, \cdots, x) > 0.$$

Define the set-valued argmax function $\phi : \Delta(\mathcal{A}) \to 2^{\Delta(\mathcal{A})}$:

$$\phi(x) := \{x' \in \Delta(\mathcal{A}) \mid U_1(x', x, \cdots, x) = \max_{x''} U_1(x'', x, \cdots, x)\}.$$

We claim that argmax function $\phi(x)$ is:

- Always non-empty and convex;

- Has a closed graph.

The first property is obvious, so we focus on the second one. Suppose that sequences $\{x_i\}$, $\{y_i\}$ satisfy $x_i \to x$, $y_i \to y$ and $y_i \in \phi(x_i)$. Since the payoff function is (Lipschitz) continuous, $\max_{x''} U_1(x'', \cdot)$ is continuous by Berge's maximum theorem. Thus $\max_{x''} U_1(x'', x_i, \cdots, x_i)$ converges to $\max_{x''} U_1(x'', x, \cdots, x)$. Meanwhile $U_1(y_i, x_i, \cdots, x_i)$ converges to $U_1(y, x, \cdots, x)$. Thus

$$U_1(y, x \cdots, x) = \lim_{i \to \infty} U_1(y_i, x_i, \cdots, x_i) = \lim_{i \to \infty} \max_{x''} U_1(x'', x_i, \cdots, x_i) = \max_{x''} U_1(x'', x, \cdots, x).$$

This implies $y \in \phi(x)$, and that $\phi$ has a closed graph. Thus by Kakutani's fixed point theorem, $\exists x^* : x^* \in \phi(x^*)$. Now we have

$$U_1(x^*, \cdots, x^*) = \max_{x''} U_1(x'', x^*, \cdots, x^*) > 0,$$

which contradicts with the assumption that the game is zero-sum and symmetric. Consequently, we prove the equation. As a result, we have

$$\min_{x_2, \cdots, x_n} \max_{x_1} U_1(x_1, \cdots, x_n) \leq \min_{x} \max_{x_1} U_1(x_1, x^{\otimes n-1}) = 0.$$

To show the second inequality can be strict, we consider a 3-player minority game. If the other two players act 0 and 1, respectively, then the learner always receive $-1/2$ payoff, which is strictly less than 0. We then finish the proofs. $\qquad\square$

## B.2 PROOF OF PROPOSITION 4.2

In this section, we will prove (2), where the inequality can be made strict in certain games.

*Proof of Proposition 4.2.* In the proof of Proposition 4.1, we have already shown that $\min_{x} \max_{x_1} U_1(x_1, x^{\otimes n-1}) = 0$. Thus, it remains to prove the inequality in (2).

Note that for any $x_1, x \in \Delta(\mathcal{A})$, we have

$$U_1(x_1, x^{\otimes n-1}) \leq \max_{x_1 \in \Delta(\mathcal{A})} U_1(x_1, x^{\otimes n-1}),$$

which implies for any $x_1 \in \Delta(\mathcal{A})$

$$\min_{x \in \Delta(\mathcal{A})} U_1(x_1, x^{\otimes n-1}) \leq \min_{x \in \Delta(\mathcal{A})} \max_{x_1 \in \Delta(\mathcal{A})} U_1(x_1, x^{\otimes n-1}).$$

By further taking maximum over $x_1 \in \Delta(\mathcal{A})$, we show that

$$\max_{x_1 \in \Delta(\mathcal{A})} \min_{x \in \Delta(\mathcal{A})} U_1(x_1, x^{\otimes n-1}) \leq \min_{x \in \Delta(\mathcal{A})} \max_{x_1 \in \Delta(\mathcal{A})} U_1(x_1, x^{\otimes n-1}).$$

To show that the inequality can be strict, we consider the scenario where the learner is involved in a 3-player majority game and plays a mixed strategy $(\beta, 1 - \beta)$ (i.e. play 0 w.p. $\beta$; play 1 w.p. $1 - \beta$). And the two opponents adopt an identical mixed strategy $(p, 1 - p)$ (i.e. play 0 w.p. $p$; play 1 w.p. $1 - p$). Then, we can calculate the payoff of the learner as $U_1(\beta, p, p) = \beta(-(1 - p)^2 + p(1 - p)) + (1 - \beta)(-p^2 + p(1 - p))$. It then follows that $\max_{\beta \in [0,1]} \min_{p \in [0,1]} U_1(\beta, p, p) \leq \max_{\beta \in [0,1]} \min_{p \in \{0,1\}} U_1(\beta, p, p) = -1/2$, which is strictly less than 0. Thus, we finish the proofs. $\qquad\square$

## B.3 PROOF OF PROPOSITION 4.3

*Proof of Proposition 4.3.* Let $\mathbb{P}^{w/o}(i_1, \ldots, i_{n-1})$ denote the probability of observing $(i_1, \ldots, i_{n-1})$ when sampling $n - 1$ points from $N$ without replacement, and let $\mathbb{P}^w(i_1, \ldots, i_{n-1})$ denote the probability of observing $(i_1, \ldots, i_{n-1})$ when sampling $n - 1$ points from $N$ with replacement. For any $a$, we then have

$$\mathbb{E}_{x_{-1}}[U_1(a, x_{-1})] = \sum_{(i_1, \ldots, i_{n-1})} \mathbb{P}^{w/o}(i_1, \ldots, i_{n-1}) U_1(a, x_{i_1}, \ldots, x_{i_{n-1}})$$

$$U_1(a, \bar{x}^{\otimes n-1}) = \sum_{(i_1, \ldots, i_{n-1})} \mathbb{P}^w(i_1, \ldots, i_{n-1}) U_1(a, x_{i_1}, \ldots, x_{i_{n-1}}).$$

Note that $\|U_1\|_{\infty} \leq 1$. Thus, we have

$$\left| \mathbb{E}_{x_{-1}}[U_1(a, x_{-1})] - U_1(a, \bar{x}^{\otimes n-1}) \right|$$

$$\leq \sum_{(i_1, \ldots, i_{n-1})} \left| \mathbb{P}^{w/o}(i_1, \ldots, i_{n-1}) - \mathbb{P}^w(i_1, \ldots, i_{n-1}) \right|$$

$$= \sum_{(i_1,\ldots,i_{n-1}) \text{ has repeated value}} \mathbb{P}^w(i_1,\ldots,i_{n-1}) - \mathbb{P}^{w/o}(i_1,\ldots,i_{n-1})$$

$$+ \sum_{(i_1,\ldots,i_{n-1}) \text{ no repeated value}} \mathbb{P}^{w/o}(i_1,\ldots,i_{n-1}) - \mathbb{P}^w(i_1,\ldots,i_{n-1})$$

$$= 2 \sum_{(i_1,\ldots,i_{n-1}) \text{ has repeated value}} \mathbb{P}^w(i_1,\ldots,i_{n-1}) - \mathbb{P}^{w/o}(i_1,\ldots,i_{n-1})$$

$$= 2\left(1 - \frac{N(N-1)\ldots(N-n+2)}{N^{n-1}}\right)$$

$$= 2\left(1 - \left(1 - \frac{1}{N}\right)\left(1 - \frac{2}{N}\right)\ldots\left(1 - \frac{n-2}{N}\right)\right)$$

$$\leq 2\left(1 - \left(1 - \frac{n-2}{N}\right)^{n-2}\right)$$

$$\leq \frac{2(n-2)^2}{N}.$$

$\square$

## C  PROOFS FOR SECTION 5

In Section C.1, we establish guarantees for the Hedge algorithm, $\text{SAOL}^{\mathcal{H}}$, and behavior cloning. In Section C.2, we provide a detailed discussion of the matching lower bounds and prove Theorem 5.4 and Theorem 5.5.

### C.1  GUARANTEES FOR EFFICIENT ALGORITHMS

In the sequel, we establish guarantees for the Hedge algorithm by proving Theorem 5.1 in Section C.1.1, for $\text{SAOL}^{\mathcal{H}}$ by proving Theorem 5.2 in Section C.1.2, and for behavior cloning by proving Theorem 5.3 in Section C.1.3.

#### C.1.1  PROOF OF THEOREM 5.1

In this section, we establish guarantees for the Hedge algorithm when facing fixed opponents.

*Proof of Theorem 5.1.* Let $a^\star \in \arg\max_{a \in \mathcal{A}} U_1(\cdot, y^{\otimes n-1})$. We then have

$$u^\star - \frac{1}{T}\sum_{t=1}^{T} u^t(x^t)$$

$$= U_1(a^\star, y^{\otimes n-1}) - \frac{1}{T}\sum_{t=1}^{T} u^t(x^t)$$

$$= \underbrace{U_1(a^\star, y^{\otimes n-1}) - \frac{1}{T}\sum_{t=1}^{T} U_1(a^\star, a^t_{-1})}_{(i)} + \underbrace{\frac{1}{T}\sum_{t=1}^{T} U_1(a^\star, a^t_{-1}) - \frac{1}{T}\sum_{t=1}^{T} U_1(x^t, a^t_{-1})}_{(ii)}$$

$$+ \underbrace{\frac{1}{T}\sum_{t=1}^{T} U_1(x^t, a^t_{-1}) - \frac{1}{T}\sum_{t=1}^{T} u^t(x^t)}_{(iii)}$$

For (i), by Hoeffding's inequality and union bound, we have with probability at least $1 - \delta$ that

$$(i) \leq O\left(\sqrt{\frac{\log(A/\delta)}{T}}\right)$$

For (ii), by Hedge algorithm, we have

$$\text{(ii)} \leq O\left(\sqrt{\frac{\log(A)}{T}}\right)$$

For (iii), note that $\{U_1(x^t, a^t_{-1}) - u^t(x^t)\}_{t=1}^T$ is a martingale difference sequence, thus by Azuma–Hoeffding inequality, we have with probability at least $1 - \delta$ that

$$\text{(iii)} \leq O\left(\sqrt{\frac{\log(1/\delta)}{T}}\right).$$

Combining the above results, we have

$$u^\star - \frac{1}{T} \sum_{t=1}^T u^t(x^t) \leq C\sqrt{\frac{\log(A/\delta)}{T}}$$

for some absolute constant $C > 0$. Thus we finish the proofs. $\qquad\square$

### C.1.2 PROOF OF THEOREM 5.2

In this section, we establish guarantees for $\text{SAOL}^{\mathcal{H}}$ when facing adaptive opponents.

The basic idea behind $\text{SAOL}^{\mathcal{H}}$ is to execute $\mathcal{H}$ in parallel over each interval within a carefully selected set. This algorithm dynamically adjusts the weight of each interval based on the previously observed regret. In each round, $\text{SAOL}^{\mathcal{H}}$ selects an interval in proportion to its assigned weight, applies $\mathcal{H}$ to each time slot within this interval, and follows its advice. Through this mechanism, $\text{SAOL}^{\mathcal{H}}$ achieves a near-optimal performance on every time interval. We will leverage the strong adaptivity of $\text{SAOL}^{\mathcal{H}}$ in our proofs.

*Proof of Theorem 5.2.* Let $\mathcal{I}$ be any fixed interval in $[0, T]$, $a_0 \in \arg\max_{a \in \mathcal{A}} \left\{ \sum_{t \in \mathcal{I}} u^t(a) \right\}$ and $u^{t,\star} := \max_{a \in \mathcal{A}} u^t(a)$. It holds that

$$\sum_{t \in \mathcal{I}} \left( u^{t,\star} - u^t(x^t) \right)$$

$$= \underbrace{\sum_{t \in \mathcal{I}} \left( u^{t,\star} - u^t(a_0) \right)}_{\text{(i)}} + \sum_{t \in \mathcal{I}} \left( u^t(a_0) - U_1(a_0, a^t_{-1}) \right)$$

$$+ \underbrace{\sum_{t \in \mathcal{I}} \left( U_1(a_0, a^t_{-1}) - U_1(x^t, a^t_{-1}) \right)}_{\text{(ii)}} + \sum_{t \in \mathcal{I}} \left( U_1(x^t, a^t_{-1}) - u^t(x^t) \right).$$

For (i), it can be seen that

$$\text{(i)} = \sum_{t \in \mathcal{I}} \left( u^{t,\star} - u^t(a_0) \right) \leq |\mathcal{I}| \max_{t \in \mathcal{I}} \left\{ u^{t,\star} - u^t(a_0) \right\} \leq 2V_{\mathcal{I}}|\mathcal{I}|.$$

Here the last inequality follows from the following argument: otherwise there exists $t_0 \in \mathcal{I}$ such that $u^{t_0,\star} - u^{t_0}(a_0) > 2V_{\mathcal{I}}$. Let $a_1 \in \arg\max_{a \in \mathcal{A}} u^{t_0}(a)$. For all $t \in \mathcal{I}$, it then holds that $u^t(a_1) \geq u^{t_0}(a_1) - V_{\mathcal{I}} = u^{t_0,\star} - V_{\mathcal{I}} > u^{t_0}(a_0) + V_{\mathcal{I}} \geq u^t(a_0)$. Contradict to the definition of $a_0$!

For (ii), we have

$$\text{(ii)} \leq \max_{a \in \mathcal{A}} \sum_{t \in \mathcal{I}} \left( U_1(a, a^t_{-1}) - U_1(x^t, a^t_{-1}) \right) \leq C(\sqrt{\log A} + \log T)\sqrt{|\mathcal{I}|},$$

where the last inequality follows from Theorem 1 in (Daniely et al., 2015).

Combining the upper bound of (i) and (ii), we have for any fixed interval $\mathcal{I} \subset [0, T]$,

$$\sum_{t \in \mathcal{I}} \left( u^{t,\star} - u^t(x^t) \right)$$

$$\leq 2V_{\mathcal{I}}|\mathcal{I}| + \sum_{t\in\mathcal{I}} \left(u^t(a_0) - U_1(a_0, a^t_{-1})\right) + C(\sqrt{\log A} + \log T)\sqrt{|\mathcal{I}|}$$
$$+ \sum_{t\in\mathcal{I}} \left(U_1(x^t, a^t_{-1}) - u^t(x^t)\right).$$

We segment the time horizon $T$ into $T/|\mathcal{I}|$ batches $\{\mathcal{I}_j\}$ with each length $|\mathcal{I}|$. It then holds for all $j$ that

$$\sum_{t\in\mathcal{I}_j} \left(u^{t,\star} - u^t(x^t)\right)$$
$$\leq 2V_{\mathcal{I}_j}|\mathcal{I}| + \sum_{t\in\mathcal{I}_j} \left(u^t(a_0) - U_1(a_0, a^t_{-1})\right) + C(\sqrt{\log A} + \log T)\sqrt{|\mathcal{I}|}$$
$$+ \sum_{t\in\mathcal{I}_j} \left(U_1(x^t, a^t_{-1}) - u^t(x^t)\right).$$

Sum over $j$ gives

$$\text{D-Reg}(T)$$
$$\leq 2V_T|\mathcal{I}| + \underbrace{\sum_{t=1}^{T} \left(u^t(a_0) - U_1(a_0, a^t_{-1})\right)}_{\text{(iii)}} + C(T/\sqrt{|\mathcal{I}|})\cdot(\sqrt{\log A} + \log T)$$
$$+ \underbrace{\sum_{t=1}^{T} \left(U_1(x^t, a^t_{-1}) - u^t(x^t)\right)}_{\text{(iv)}}.$$

For (iii), note that $\{u^t(a) - U_1(a, a^t_{-1})\}_{t=1}^{T}$ is a martingale difference sequence, we have with probability at least $1 - \delta$ that

$$\text{(iii)} \leq \max_{a\in\mathcal{A}} \sum_{t=1}^{T} \left(u^t(a) - U_1(a, a^t_{-1})\right) \leq O\left(\sqrt{T\log(A/\delta)}\right),$$

where the last inequality follows from Azuma–Hoeffding inequality and union bound.

For (iv), note that $\{U_1(x^t, a^t_{-1}) - u^t(x^t)\}_{t=1}^{T}$ is a martingale difference sequence, thus by Azuma–Hoeffding inequality, we have with probability at least $1 - \delta$ that

$$\text{(iv)} \leq O\left(\sqrt{T\log(1/\delta)}\right).$$

Consequently we have with probability at least $1 - \delta$ that

$$\text{D-Reg}(T) \leq 2V_T|\mathcal{I}| + C(T/\sqrt{|\mathcal{I}|})\cdot(\sqrt{\log A} + \log T) + O\left(\sqrt{T\log(A/\delta)}\right).$$

Choosing $|\mathcal{I}| = (T/V_T)^{2/3}$, we have with probability at least $1 - \delta$ that

$$\text{D-Reg}(T) \leq O\left(V_T^{1/3} T^{2/3}(\sqrt{\log(A/\delta)} + \log T)\right).$$

Finally, by the definition of $u_{\text{avg}}(T)$ and $u^\dagger$, we show that

$$u_{\text{avg}}(T) \geq u^\dagger - CV_T^{1/3} T^{-1/3}\left(\sqrt{\log(A/\delta)} + \log T\right)$$

for some absolute constant $C$.

$\square$

---

**Algorithm 2** Behavior Cloning

---

1: In the first round, play $a \sim \text{Uniform}(\mathcal{A})$.
2: **for** $t = 2, \ldots, T$ **do**
3:    Play $a_2^{t-1}$, i.e. the action played by Player 2 in the last round.

---

### C.1.3 PROOF OF THEOREM 5.3

In this section, we establish guarantees for behavior cloning when facing adaptive opponents.

*Proof of Theorem 5.3.* Note that

$$
\mathbb{E}\left[\sum_{t=1}^{T} u^t(x^t)\right] \geq -1 + \mathbb{E}\left[\sum_{t=2}^{T} u^t(x^t)\right]
$$

$$
= -1 + \mathbb{E}\left[\sum_{t=2}^{T} U_1(a_2^{t-1}, (y^t)^{\otimes n-1})\right]
$$

$$
\geq -1 - V_T - \mathbb{E}\left[\sum_{t=2}^{T} U_1(a_2^{t-1}, (y^{t-1})^{\otimes n-1})\right] \qquad \text{(by the defition of } V_T)
$$

$$
= -1 - V_T - \mathbb{E}\left[\sum_{t=2}^{T} U_1(y^{t-1}, (y^{t-1})^{\otimes n-1})\right] \qquad \text{(since } a_2^{t-1} \sim y^{t-1})
$$

$$
= -1 - V_T \qquad \text{(since the game is symmetric and zero-sum)}
$$

Finally, by the definition of $u_{\text{avg}}(T)$, we finish the proofs. □

### C.2 MATCHING LOWER BOUNDS

Upon examining Theorem 5.2 alongside Theorem 5.3, it becomes apparent that Theorem 5.2 benchmarks against a more stringent standard (i.e., the dynamic oracle) and incurs a larger error of $V_T^{1/3}T^{-1/3}$, while Theorem 5.3 sets its comparison against a baseline metric (i.e., the average payoff) and attains a smaller error of $V_T/T$. Regarding this observation, one might aspire to devise an algorithm whose payoff satisfies: $u_{\text{avg}}(T) \geq u^\dagger - \tilde{O}(V_T/T)$. However, Theorem 5.4 and Theorem 5.5 demonstrate that such a goal is unattainable, by exploring the fundamental limits faced when competing against non-stationary opponents.

Theorem 5.4 shows, when contending with non-stationary opponent, the optimal algorithm must incur a dynamic regret at least order of $V_T^{1/3}T^{2/3}$, closing off the possibility of attaining a better $V_T$ rate. It's noteworthy that a similar lower bound for dynamic regret has already been established under broader conditions Besbes et al. (2014). The distinction of Theorem 5.4 lies in further restricting the hard problems to be symmetric games, implying that the structure of symmetric game does not offer an advantage in improving dynamic regret in the worst case. By comparing this lower bound with Theorem 5.2, it is evident that $\text{SAOL}^{\mathcal{H}}$ is demonstrated to be minimax optimal, albeit with the inclusion of some logarithmic factors.

Theorem 5.5 establishes the fundamental limit when comparing to average payoff 0. The guarantees achieved by Theorem 5.3 can not be improved in the worst case, showing behavior cloning is demonstrated to be optimal upto some constant.

### C.2.1 PROOF OF THEOREM 5.4

*Proof of Theorem 5.4.* We define

$$
U_1^{(3)}(a, b, c) := \begin{cases} \text{payoff for 3-player majority game} & \text{if } a, b, c \in \{0, 1\} \\ -1 & \text{if } a \notin \{0, 1\}, b, c \in \{0, 1\} \\ \text{defined by symmetric} & \text{o.w.} \end{cases}
$$

which is basically the payoff function for 3-player majority game with extra dummy actions. We then define

$$U_1^{(n)}(a, a_2, \ldots, a_n) := \frac{1}{(n-1)(n-2)} \sum_{2 \leq i \neq j \leq n} U_1^{(3)}(a, a_i, a_j).$$

We consider a game that evolves stochastically, with $n$ players, action space $\mathcal{A} = \{0, 1, \ldots, A-1\}$, and the payoff function of the first player given by $U_1^{(n)}$. We segment the decision horizon $T$ into $T/\Delta_T$ batches $\{\mathcal{T}_j\}$, with each batch comprising $\Delta_T$ episodes. We consider two distinct scenarios:

- Case1: All the other players employ a mixture strategy $(1/2 - \epsilon, 1/2 + \epsilon)$ (i.e., playing 0 with probability $1/2 - \epsilon$, playing 1 with probability $1/2 + \epsilon$);

- Case 2: All the other players employ a mixture strategy $(1/2 + \epsilon, 1/2 - \epsilon)$ (i.e., playing 0 with probability $1/2 + \epsilon$, playing 1 with probability $1/2 - \epsilon$);

At the beginning of each batch, one of these scenarios is randomly selected (with equal probability) and remains constant throughout that batch.

Let $m = T/\Delta_T$ represent total number of batches. We fix some algorithm and a batch $j \in \{1, \ldots, m\}$. Let $\delta_j \in \{1, 2\}$ indicate batch $j$ belongs to Case1 or Case2. We denote by $\mathbb{P}_{\delta_j}^j$ the probability distribution conditioned on batch $j$ belongs to Case $\delta_j$, and by $\mathbb{P}_0$ the probability distribution when all the other players employ a mixture strategy $(1/2, 1/2)$. We further denote by $\mathbb{E}_{\delta_j}^j[\cdot]$ and $\mathbb{E}_0[\cdot]$ the corresponding expectations. We denote by $N_a^j$ the number of times action $a$ was played in batch $j$. If the batch $j$ belongs to Case $\delta_j$, then the optimal action in the batch is $-\delta_j + 2$. We first present a useful lemma.

**Lemma C.1.** *Let* $f : \{-1, 0, 1/2\}^{|\mathcal{T}_j| \times A} \to [0, M]$ *be any bounded real function defined on the payoff matrices $R$. Then, for any $\delta_j \in \{1, 2\}$, $\epsilon \leq 1/4$:*

$$\mathbb{E}_{\delta_j}^j[f(R)] - \mathbb{E}_0[f(R)] \leq \frac{M}{2}\sqrt{-2|\mathcal{T}_j|\ln(1 - 4\epsilon^2)} \leq 2M\epsilon\sqrt{\Delta_T}.$$

By Lemma C.1 with $f = N_{-\delta_j+2}^j$, we have

$$\mathbb{E}_{\delta_j}^j[N_{-\delta_j+2}^j] - \mathbb{E}_0[N_{-\delta_j+2}^j] \leq 2\epsilon|\mathcal{T}_j|\sqrt{\Delta_T}. \tag{4}$$

Note that

$$\mathbb{E}_{\delta_j}^j[u^t(x^t)] = -\mathbb{P}_{\delta_j}^j(x^t \notin \{0, 1\}) + (-\epsilon - 2\epsilon^2)\mathbb{P}_{\delta_j}^j(x^t = \delta_j - 1) + (\epsilon - 2\epsilon^2)\mathbb{P}_{\delta_j}^j(x^t = -\delta_j + 2)$$

$$\leq (-\epsilon - 2\epsilon^2)\mathbb{P}_{\delta_j}^j(x^t \neq -\delta_j + 2) + (\epsilon - 2\epsilon^2)\mathbb{P}_{\delta_j}^j(x^t = -\delta_j + 2)$$

$$= -\epsilon - 2\epsilon^2 + 2\epsilon \cdot \mathbb{P}_{\delta_j}^j(x^t = -\delta_j + 2),$$

therefore,

$$\mathbb{E}_{\delta_j}^j\left[\sum_{t \in \mathcal{T}_j} u^t(x^t)\right] \leq (-\epsilon - 2\epsilon^2)|\mathcal{T}_j| + 2\epsilon \cdot \mathbb{E}_{\delta_j}^j[N_{-\delta_j+2}^j]$$

$$\leq (-\epsilon - 2\epsilon^2)|\mathcal{T}_j| + 2\epsilon \cdot \mathbb{E}_0^j[N_{-\delta_j+2}^j] + 4\epsilon^2|\mathcal{T}_j|\sqrt{\Delta_T}. \tag{by (4)}$$

Consequently, we have

$$\frac{1}{2}\mathbb{E}_1^j\left[\sum_{t \in \mathcal{T}_j} u^t(x^t)\right] + \frac{1}{2}\mathbb{E}_2^j\left[\sum_{t \in \mathcal{T}_j} u^t(x^t)\right] \leq (-\epsilon - 2\epsilon^2)|\mathcal{T}_j| + \epsilon|\mathcal{T}_j| + 4\epsilon^2|\mathcal{T}_j|\sqrt{\Delta_T}. \tag{5}$$

It then holds that

$$\mathbb{E}_{\text{Alg}}\left[\sum_{j=1}^m \sum_{t \in \mathcal{T}_j} u^t(x^t)\right] = \sum_{j=1}^m \mathbb{E}_{\text{Alg}}\left[\sum_{t \in \mathcal{T}_j} u^t(x^t)\right]$$

$$= \sum_{j=1}^m \mathbb{E}_{\text{Alg}}\left[\frac{1}{2}\mathbb{E}_1^j\left[\sum_{t\in\mathcal{T}_j} u^t(x^t)\right] + \frac{1}{2}\mathbb{E}_2^j\left[\sum_{t\in\mathcal{T}_j} u^t(x^t)\right]\right]$$

$$\leq \sum_{j=1}^m ((-\epsilon - 2\epsilon^2)|\mathcal{T}_j| + \epsilon|\mathcal{T}_j| + 4\epsilon^2|\mathcal{T}_j|\sqrt{\Delta_T})$$

$$= -2\epsilon^2 T + 4\epsilon^2 T\sqrt{\Delta_T}.$$

Set $\epsilon = \min\{1/(8\sqrt{\Delta_T}), V_T\Delta_T/T\}$. We then have

$$\mathbb{E}_{\text{Alg}}[\text{D-Reg}(T)] = (\epsilon - 2\epsilon^2)T - \mathbb{E}_{\text{Alg}}\left[\sum_{t=1}^T u^t(x^t)\right]$$

$$\geq (\epsilon - 2\epsilon^2)T - (-2\epsilon^2 T + 4\epsilon^2 T\sqrt{\Delta_T})$$

$$= \epsilon T - 4\epsilon^2 T\sqrt{\Delta_T}$$

$$= \epsilon T(1 - 4\epsilon\sqrt{\Delta_T})$$

$$\geq \frac{1}{2}\epsilon T$$

$$= \frac{1}{2}\min\left\{\frac{1}{8\sqrt{\Delta_T}}, \frac{V_T\Delta_T}{T}\right\}T.$$

Choosing $\Delta_T = (T/V_T)^{2/3}$, we then have

$$\mathbb{E}_{\text{Alg}}[\text{D-Reg}(T)] \geq C V_T^{1/3} T^{2/3}.$$

Recall the definition of $u_{\text{avg}}(T)$ and $u^\dagger$, we then finish the proofs. □

We prove Lemma C.1 in the following.

*Proof of Lemma C.1.* We have that

$$\mathbb{E}_{\delta_j}^j[f(R)] - \mathbb{E}_0[f(R)] = \sum_R f(R)\left(\mathbb{P}_{\delta_j}^j(R) - \mathbb{P}_0(R)\right)$$

$$\leq \sum_{R:\mathbb{P}_{\delta_j}^j(R)\geq\mathbb{P}_0(R)} f(R)\left(\mathbb{P}_{\delta_j}^j(R) - \mathbb{P}_0(R)\right)$$

$$\leq M\sum_{R:\mathbb{P}_{\delta_j}^j(R)\geq\mathbb{P}_0(R)}\left(\mathbb{P}_{\delta_j}^j(R) - \mathbb{P}_0(R)\right)$$

$$= \frac{M}{2}\|\mathbb{P}_{\delta_j}^j - \mathbb{P}_0\|_{\text{TV}}$$

$$\leq \frac{M}{2}\sqrt{2\text{KL}(\mathbb{P}_0 \parallel \mathbb{P}_{\delta_j}^j)}, \tag{6}$$

where the last ineqaulity follows from Pinsker's inequality. Let $R_t \in \mathbb{R}^A$ be a random vector denoting the payoff for each action at time $t$, and let $R^t \in \mathbb{R}^{t\times A}$ denote the payoff matrix received upon time $t$: $R^t = [R_1, \ldots, R_t]^T$. By the chain rule for the relative entropy, we have

$$\text{KL}(\mathbb{P}_0 \parallel \mathbb{P}_{\delta_j}^j) = \sum_{t=1}^{|\mathcal{T}_j|} \mathbb{E}_{R^{t-1}}\left[\text{KL}\left(\mathbb{P}_0(R_t \mid R^{t-1}) \parallel \mathbb{P}_{\delta_j}^j(R_t \mid R^{t-1})\right)\right]. \tag{7}$$

Note that

$$\mathbb{P}_0(R_t = [-1, 0, -1, \ldots, -1] \mid R^{t-1}) = \mathbb{P}_0(R_t = [0, -1, -1, \ldots, -1] \mid R^{t-1}) = 1/4$$

$$\mathbb{P}_0(R_t = [1/2, 1/2, -1, \ldots, -1] \mid R^{t-1}) = 1/2.$$

In the case $\delta_j = 1$, we have

$$\mathbb{P}^j_{\delta_j}(R_t = [-1, 0, -1, \ldots, -1] \mid R^{t-1}) = (1/2 + \epsilon)^2$$

$$\mathbb{P}^j_{\delta_j}(R_t = [0, -1, -1, \ldots, -1] \mid R^{t-1}) = (1/2 - \epsilon)^2$$

$$\mathbb{P}^j_{\delta_j}(R_t = [1/2, 1/2, -1, \ldots, -1] \mid R^{t-1}) = 2(1/2 + \epsilon)(1/2 - \epsilon).$$

In the case $\delta_j = 2$, we have

$$\mathbb{P}^j_{\delta_j}(R_t = [-1, 0, -1, \ldots, -1] \mid R^{t-1}) = (1/2 - \epsilon)^2$$

$$\mathbb{P}^j_{\delta_j}(R_t = [0, -1, -1, \ldots, -1] \mid R^{t-1}) = (1/2 + \epsilon)^2$$

$$\mathbb{P}^j_{\delta_j}(R_t = [1/2, 1/2, -1, \ldots, -1] \mid R^{t-1}) = 2(1/2 + \epsilon)(1/2 - \epsilon).$$

Thus, we have

$$\mathsf{KL}\left(\mathbb{P}_0(R_t \mid R^{t-1}) \,\|\, \mathbb{P}^j_{\delta_j}(R_t \mid R^{t-1})\right) \tag{8}$$

$$= \frac{1}{4} \ln \frac{1/4}{(1/2 + \epsilon)^2} + \frac{1}{4} \ln \frac{1/4}{(1/2 - \epsilon)^2} + \frac{1}{2} \ln \frac{1/2}{2(1/2 + \epsilon)(1/2 - \epsilon)}$$

$$= -\ln\left(1 - 4\epsilon^2\right). \tag{9}$$

Combining (6), (7) and (8), we have

$$\mathbb{E}^j_{\delta_j}[f(R)] - \mathbb{E}_0[f(R)] \le \frac{M}{2}\sqrt{-2|\mathcal{T}_j|\ln\left(1 - 4\epsilon^2\right)}.$$

If we further have $\epsilon \le 1/4$, it then holds that $-\ln\left(1 - 4\epsilon^2\right) \le 16\ln(4/3)\epsilon^2$ and consequently

$$\mathbb{E}^j_{\delta_j}[f(R)] - \mathbb{E}_0[f(R)] \le \frac{M}{2}\sqrt{-2|\mathcal{T}_j|\ln\left(1 - 4\epsilon^2\right)} \le 2M\epsilon\sqrt{|\mathcal{T}_j|} \le 2M\epsilon\sqrt{\Delta_T}.$$

$\square$

### C.2.2    PROOF OF THEOREM 5.5

*Proof of Theorem 5.5.* We consider a game that evolves stochastically, with $n$ players, action space $\mathcal{A} = \{0, 1, \ldots, A - 1\}$, and the same payoff function $U_1^{(n)}$ as outlined in Theorem 5.4. We segment the decision horizon $T$ into $T/\Delta_T$ batches $\{\mathcal{T}_j\}$, with each batch comprising $\Delta_T$ episodes. We consider two distinct scenarios:

- Case1: All the other players play 0;

- Case 2: All the other players play 1.

In Case 1, we have $u^t(0) = 0$ and $u^t(a) = -1$ for all $a \ne 0$. In Case 2, we have $u^t(1) = 0$ and $u^t(a) = -1$ for all $a \ne 1$. At the beginning of each batch, one of these scenarios is randomly selected (with equal probability) and remains constant throughout that batch.

Let $m = T/\Delta_T$ represent total number of batches. We fix some algorithm. Let $\delta_j \in \{1, 2\}$ indicate batch $j$ belongs to Case1 or Case2. We denote by $\mathbb{P}^j_{\delta_j}$ the probability distribution conditioned on batch $j$ belongs to Case $\delta_j$, and by $\mathbb{E}^j_{\delta_j}[\cdot]$ the corresponding expectation. It then holds that

$$\mathbb{E}_{\text{Alg}}\left[\sum_{j=1}^m \sum_{t \in \mathcal{T}_j} u^t(x^t)\right] = \sum_{j=1}^m \mathbb{E}_{\text{Alg}}\left[\sum_{t \in \mathcal{T}_j} u^t(x^t)\right]$$

$$= \sum_{j=1}^m \mathbb{E}_{\text{Alg}}\left[\frac{1}{2}\mathbb{E}_1^j\left[\sum_{t \in \mathcal{T}_j} u^t(x^t)\right] + \frac{1}{2}\mathbb{E}_2^j\left[\sum_{t \in \mathcal{T}_j} u^t(x^t)\right]\right]$$

$$\leq \sum_{j=1}^{m} \mathbb{E}_{\text{Alg}} \left[ \frac{1}{2} \mathbb{E}_1^j \left[ u^{t^{j,1}}(x^{t^{j,1}}) \right] + \frac{1}{2} \mathbb{E}_2^j \left[ u^{t^{j,1}}(x^{t^{j,1}}) \right] \right],$$

where $t^{j,1}$ represents the first episode of batch $j$ and the inequality follows from the fact that $u^t \leq 0$. Note that

$$\frac{1}{2} \mathbb{E}_1^j \left[ u^{t^{j,1}}(x^{t^{j,1}}) \right] + \frac{1}{2} \mathbb{E}_2^j \left[ u^{t^{j,1}}(x^{t^{j,1}}) \right] = -\frac{1}{2} \left( \mathbb{P}_1^j(x^{t^{j,1}} \neq 0) + \mathbb{P}_2^j(x^{t^{j,1}} \neq 1) \right)$$

$$= -\frac{1}{2} \left( \mathbb{P}(x^{t^{j,1}} \neq 0) + \mathbb{P}(x^{t^{j,1}} \neq 1) \right) \leq -\frac{1}{2},$$

where the second equation follows from the fact that $x^{t^{j,1}}$ is independent of $\delta_j$. Thus, we have

$$\mathbb{E}_{\text{Alg}} \left[ \sum_{j=1}^{m} \sum_{t \in \mathcal{T}_j} u^t(x^t) \right] \leq -\frac{m}{2} = -\frac{T}{2\Delta_T}.$$

Choosing $\Delta_T = T/V_T$, we have

$$\mathbb{E}_{\text{Alg}} \left[ \sum_{j=1}^{m} \sum_{t \in \mathcal{T}_j} u^t(x^t) \right] \leq -V_T/2.$$

Recall the definition of $u_{\text{avg}}(T)$, we then finish the proofs. $\qquad\square$

## D  EXPERIMENTS DETAILS

In this section, we provide additional details for our experiments.

### D.1  ALGORITHMS

We refer readers to Algorithm 3-6 for detailed implementation of algorithms in the experiment. For **MV**, we choose $\eta = 1$. For **SDG**, we choose $\eta = 2$.

---

**Algorithm 3** Self-Play

---

**Require:** Number of iterations $T$, action space $\mathcal{A}$, learning rate $\eta_t = \eta \sqrt{\frac{\log |\mathcal{A}|}{t}}$, number of players
    $n$, and initialize strategy $x_0$.
1: **for** t = 1 to T **do**
2:    Sample actions $a_i^{t-1} \sim x_{t-1}$ for $i = 2, \ldots, n$. Denote $a_{-1}^{t-1} := (a_2^{t-1}, \ldots, a_n^{t-1})$.
3:    Update
$$x_t(a) \propto x_{t-1}(a) \exp\{\eta_t U_1(a, a_{-1}^{t-1})\}, \forall a \in \mathcal{A}.$$

---

**Algorithm 4** Self-Play with Regularization

---

**Require:** Number of iterations $T$, action space $\mathcal{A}$, learning rate $\eta_t = \eta \sqrt{\frac{\log |\mathcal{A}|}{t}}$, number of players
    $n$, initialize strategy $x_0$, meta-strategy $y_{\text{meta}}$, and regularization parameter $\lambda$.
1: **for** t = 1 to T **do**
2:    Sample actions $a_i^{t-1} \sim x_{t-1}$ for $i = 2, \ldots, n$. Denote $a_{-1}^{t-1} := (a_2^{t-1}, \ldots, a_n^{t-1})$.
3:    Update
$$x_t(a) \propto \exp\left\{ \frac{\log x_0(a) + \sum_{\tau < t} \eta_\tau U_1(a, a_{-1}^\tau) + \lambda \sum_{\tau < t} \eta_\tau \log y_{\text{meta}}(a)}{1 + \lambda \sum_{\tau < t} \eta_\tau} \right\}, \forall a \in \mathcal{A}.$$

---

---

**Algorithm 5** Hedge

---

**Require:** Number of iterations $T$, action space $\mathcal{A}$, learning rate $\eta_t = \eta\sqrt{\frac{\log |\mathcal{A}|}{t}}$, number of players $n$, and initialize strategy $x_0$.

1: **for** t = 1 to T **do**
2:    Sample actions $a_i^{t-1} \sim y_{\text{meta}}$ for $i = 2, \ldots, n$. Denote $a_{-1}^{t-1} := (a_2^{t-1}, \ldots, a_n^{t-1})$.
3:    Update
$$x_t(a) \propto x_{t-1}(a)\exp\{\eta_t U_1(a, a_{-1}^{t-1})\}, \forall a \in \mathcal{A}.$$

---

**Algorithm 6** Exploiter for strategy $x$

---

**Require:** Number of iterations $T$, action space $\mathcal{A}$, learning rate $\eta_t = \eta\sqrt{\frac{\log |\mathcal{A}|}{t}}$, number of players $n$, and initialize strategy $x_0$.

1: **for** t = 1 to T **do**
2:    Sample actions $a_1^{t-1} \sim x$.
3:    Sample actions $a_i^{t-1} \sim x_{t-1}$ for $i = 2, \ldots, n$. Denote $a_{-1}^{t-1} := (a_2^{t-1}, \ldots, a_n^{t-1})$.
4:    Update

$$x_t(a) \propto x_{t-1}(a)\exp\left\{-\frac{\eta_t}{n-1}\sum_{i=2}^{n} U_1\left(a_1^{t-1}, a_{-1}^{t-1}[: i-1], a, a_{-1}^{t-1}[i+1 :]\right)\right\}, \forall a \in \mathcal{A}.$$

---

### D.2 COMPUTATION RESOURCES

The experiments are conducted on a server with 256 CPUs. Each experiment can be completed in a few minutes.

