# OpenReview forum: "Securing Equal Share: A Principled Approach for Learning Multiplayer Symmetric Games"
_ICLR.cc/2025/Conference — Submitted to ICLR 2025_

### Official Review · Reviewer_Yqqg · 2024-10-29

**Soundness:** 3
**Presentation:** 2
**Contribution:** 2
**Rating:** 5
**Confidence:** 4

**Summary:**

This paper examines multiplayer symmetric constant-sum games with more than two players in a competitive setting. In contrast to two-player zero-sum games, equilibria in multiplayer games are neither unique nor non-exploitable, failing to provide meaningful guarantees when competing against opponents who play different equilibria or non-equilibrium strategies. This gives rise to a series of long-lasting fundamental questions in multiplayer games regarding suitable objectives, solution concepts, and principled algorithms. This paper addresses these challenges by focusing on the minmax strategy and  the natural objective of equal share—securing an expected payoff of C/n in an n-player symmetric game with a total payoff of C. They rigorously identify the theoretical conditions under which achieving an equal share is tractable and design a series of efficient algorithms, inspired by no-regret learning, that provably attain approximate equal share across various settings.

**Strengths:**

They rigorously identify the theoretical conditions under which achieving an equal share is tractable.

**Weaknesses:**

The significance of the contribution is limited.

1) The so-called suitable solution concept for learning in multiplayer games is a minimax strategy, which is not new.

2) This paper mentioned that symmetric games are popular in practice, however, the large games mentioned in this paper are not symmetric. Please give more detailed real-world cases.

3) This paper mentioned that all asymmetric games can be converted to symmetric games. More details should be given, especially how the results in this paper can be used in all asymmetric games.

4) This paper shows that the equal share can be secured under two conditions, making a multiplayer game equivalent to a two-player zero-sum game with the opponent fixed.  This case is trivial for securing an equal share. As two-player zero-sum games have been studied well, then the following algorithm is trivial.

These results may not help solve general multiplayer games.

**Questions:**

no.

---

> ### Author Response · Authors · 2024-11-21
>
> Thanks a lot for reading our paper and for your insightful comments.
>
> 1.**The so-called suitable solution concept for learning in multiplayer games is a minimax strategy, which is not new.**
>
> We would like to clarify that our proposed solution concept, $\min_{x} \max_{x_1} U_1(x_1, x^{\otimes n-1})$, is indeed new and fundamentally different from the conventional objectives in multiplayer game settings.
> This formulation is meaningful only through our unique perspective of achieving an equal share.
> Traditional analyses typically focus on the solution $\max_{x_1} \min_{x_2, \cdots, x_n} U_1(x_1, \cdots, x_n)$, which, as the reviewer notes, is not novel. However, as shown in Proposition 4.1, this conventional approach does not yield the correct solution concept required for our setting (i.e., achieving equal share). Our results in Propositions 4.1 and 4.2 demonstrate that out of the four minimax/maximin formulations we examined, only our proposed solution, $\min_{x} \max_{x_1} U_1(x_1, x^{\otimes n-1}) $, successfully achieves an equal share in multiplayer symmetric games. The other three formulations can yield values strictly less than zero under specific game conditions, further highlighting the fundamental differences between two-player zero-sum games and multiplayer symmetric games.
>
> 2.**The large games mentioned in this paper are not symmetric. Why all asymmetric games can be converted to symmetric games?**
>
> We appreciate the reviewer’s observations and agree that games like Mahjong, Poker, Avalon, and Diplomacy are not strictly symmetric because the positions or roles assigned to players may influence their strategic options. However, as noted in footnote 2, we can effectively transform these games into symmetric ones by introducing an initial stage where each player’s position is randomly assigned. This randomization process effectively symmetrizes the game, as each player has an equal probability of assuming any role, ensuring that no player has a consistent positional advantage. Therefore, under this transformed setting, our analysis remains applicable.
>
> To further clarify the mathematical transformation, we enlarge the action space to incorporate both the player's position and their actions. Specifically, for each player $i$, we define $(P_i, a_i)$ where $P_i$ represents the player's position or role in the game, and $a_i$ denotes their action. We then define a symmetric utility function $U^{sym}_i$ as follows:
>
> $U^{sym}_i((P_1, a_1), \ldots, (P_n, a_n))$
>
> $:=U^{asym}\_{\sigma^{-1}(i)}(a_{\sigma^{-1}(1)}, \ldots, a_{\sigma^{-1}(n)}),$
>
>
> where $\sigma$ is a permutation that reorders player positions such that $\sigma(P_1, \ldots, P_n) = (1, \ldots, n)$.
> The action space is defined as $\bigcup_{j=1}^n (P_j \times \mathcal{A}_j) $, where $ \mathcal{A}_j $ denotes the set of actions available to players when they are in position or role $P_j $.
> This transformation aligns the players' positions with a standardized symmetric framework, enabling our analysis to apply to games that are asymmetric by nature but can be symmetrized under random initial positioning.
>
>
> 3.**This paper shows that the equal share can be secured under two conditions, making a multiplayer game equivalent to a two-player zero-sum game with the opponent fixed.  Two-player zero-sum games have been studied well.**
>
> We agree with the reviewer that a multiplayer symmetric game can be viewed as equivalent to a two-player zero-sum game under two conditions. However, traditional results from two-player zero-sum games no longer apply directly, as the unique structure of this setting—all opponents sharing the same policy—violates the convexity assumption of the minimax theorem. In Proposition 4.2, we provide a counterexample that contradicts these traditional results, demonstrating that $\max_{x_1} \min_{x} U_1$ can be strictly less than zero, rather than equal to zero, as conventional theory would suggest.
>
> Additionally, we extend our analysis to the adaptive opponent setting in Section 5.2, providing further theoretical insights that strengthen our contributions.
> Please also refer to our general response for the discussion on the technical contribution.
>
> 4.**These results may not help solve general multiplayer games.**
>
> The challenge of solving general multiplayer games is widely recognized as a difficult problem, and addressing it fully is beyond the scope of this work. However, it would be unfair to overlook our contributions simply because they do not solve this extremely complex problem in its entirety. Progress in any field is made incrementally, with major advances often built upon understanding and solving more tractable subproblems. We see our work as an important initial step in establishing principled methodologies and algorithms that can guide future research in multiplayer games.
>
> For further details regarding the novelty and technical contributions of this paper, please refer to our general response.

---

> > ### Comment · Reviewer_Yqqg · 2024-11-25
> >
> > Thanks for your rebuttal.

---

> > > ### Author Response · Authors · 2024-11-25
> > >
> > > Thank you once again for taking the time to carefully consider our rebuttal and engage in the discussion. We truly value your thoughtful feedback, which has helped us clarify our work. As the discussion period comes to a close, we sincerely hope that our responses have sufficiently addressed your concerns. If so, we would be deeply grateful if you could kindly reconsider your evaluation and adjust your score accordingly. Of course, we remain available to address any further questions or concerns you may have in the remaining time.

---

### Official Review · Reviewer_5Z1z · 2024-11-03

**Soundness:** 3
**Presentation:** 3
**Contribution:** 2
**Rating:** 5
**Confidence:** 4

**Summary:**

The paper considers multi-player symmetric zero-sum games, a class which captures many games from realistic applications. In light of the deficiencies of existing solution concepts in such games, the paper proposes a new solution concept, namely equal share---a player must secure a utility or 0. They proceed by identifying sufficient and necessary conditions under which equal share is attainable. Experiments are also conducted to highlight some of the theoretical results.

**Strengths:**

Identifying reasonable solution concepts beyond two-player zero-sum games has been a major challenge in game theory. Common solution concepts, such as Nash equilibria and correlated equilibria, have many deficiencies, and so attempting to explore different concepts that overcome those issues is certainly a fruitful research direction. The paper contributes to this line of work by proposing a natural objective, described above, which appears to be novel in its current form. Further, the authors provide necessary and sufficient conditions under which that solution concept can be attained. The conditions are very interpretable and natural, and provide a theoretical framework to design principled algorithms in practice. It is plausible that the approach of the paper can guide the performance of systems in practice. The paper is generally well-written, and all claims appear to be sound. The scope of the paper would also make it a good fit for a conference such as ICLR.

**Weaknesses:**

On the negative side, my main concern is about the novelty of the results. First, all the observations made in the paper (Section 3 and in part Section 6) about the other solution concepts and the self-play framework are well-known and immediate. It is clear that one can devise simple examples where common solution concepts fail in terms of the objective put forward by the paper. The more interesting question, in my opinion, is why the self-play framework performs that well in games such as multi-player poker despite those obvious deficiencies. In terms of the conditions identified in the paper in order to achieve equal share, it is again fairly immediate to see that there are necessary and sufficient. In particular, when the opponents are not fully adaptive, most of the positive results obtained in Section 5 follow readily from existing results in the literature on dynamic regret. As to whether equal share is a reasonable objective, I don't believe there is a definite answer. The fact that it does not correspond to an equilibrium is an obvious concern, but I understand that going beyond equilibrium will be necessary to make progress in this line of work. Overall, I believe that the technical contributions of the paper do not go far enough compared to existing results to merit acceptance.

**Questions:**

I have no further questions.

---

> ### Author Response · Authors · 2024-11-21
>
> Thanks a lot for reading our paper and for your insightful comments.
>
> 1.**Novelty of the results and the technical contribution.**
> Please refer to our general response.
>
> 2.**The Success of Self-Play Frameworks.**
>
> The primary focus of this paper is to explore how to approach multiplayer games in a principled manner. We use a bottom-up approach, which identifies a suitable objective that extends beyond equilibria---namely, achieving an equal share of rewards---and develop corresponding solution concepts and efficient algorithms within a comprehensive framework.
>
> While the achievements of self-play algorithms in games like multi-player poker are undeniably impressive, it is important to note that these successes are often closely tied to specific contexts and rely heavily on extensive engineering optimizations, often with significant input from human expertise. We agree that examining why these algorithms succeed and whether they can serve as general-purpose solutions across a wide range of game settings is a very compelling future question, which is beyond the scope of this current paper.

---

> > ### Author Response · Authors · 2024-11-25
> >
> > Thank you for taking the time to review our paper and provide your thoughtful feedback. We hope our response has addressed your concerns and clarified any uncertainties. As the discussion period is nearing its conclusion, please feel free to share any remaining questions or comments. We would be happy to discuss them further.

---

> > > ### Comment · Reviewer_5Z1z · 2024-11-26
> > >
> > > I thank the authors for their detailed feedback. I maintain most of my original concerns. Although the paper has certain merits, I don't believe the results go far enough compared to the existing literature.

---

### Official Review · Reviewer_sS4W · 2024-11-04

**Soundness:** 4
**Presentation:** 2
**Contribution:** 3
**Rating:** 6
**Confidence:** 3

**Summary:**

The paper studies n-player symmetric constant-sum games, specifically how an agent can guarantee themselves a 1/n- fraction, i.e. "equal share" of the total payoff.  They identify conditions under which a learner can guarantee themselves this equal share against (n-1) opponents. The paper also studies various no-regret algorithms that can guarantee equal share under these conditions and compares their performance to self-play based meta-learners.

They identify two conditions necessary for a single player, called the learner, to learn a strategy that gives them their equal share in a repeated interaction -
1. All the other (n-1) players, called the opponents, must deploy the same strategy
2. The opponents must have limited adaptivity over the rounds of play

The first condition is shown via analyzing the one-shot version of the game and showing that the minimax theorem does not hold in symmetric n-player constant sum games. This condition is justified by studying the game against opponents drawn from a large random population. The second condition is additionally required to show that strongly adaptive no-regret algorithms achieve small dynamic regret against opponents with limited adaptivity. For intermediate rates of adaptivity, the paper also studies the payoff obtained by mimicking the identical opponents. The final theoretical result is to show matching lower bounds for dynamic regret.

The paper also experimentally compares the performance of these algorithms against that of existing self-play based frameworks.

**Strengths:**

The main strength of the paper is coming up with a simple and compelling framework to study learning in n-player symmetric games. The paper has some clean results showing that the minimax theorem and variants do not hold in multiplayer symmetric games and uses this to place some reasonable restrictions on opponent behavior. In particular, the paper introduces some exemplar multiplayer symmetric games, such as the majority/ minority vote game, which demonstrate useful properties that provide insight into the nature of equilibria and best-responses in multiplayer symmetric games. The results related to these parts

Another strength of the paper is in coming up with a natural setting (where opponents are drawn from a large population) to justify working with the condition that all opponents behave identically.

The paper establishes matching upper and lower bounds for payoffs in the online learning setting -- for different regimes based on how swiftly the opponent's strategy and the resulting environment is changing. In particular, the lower bounds extend previous results about general multiplayer games to the special case of symmetric games.

**Weaknesses:**

There are two main weaknesses in my view:

1. One of the main technical results in the paper is about dynamic regret bounds against opponents with limited adaptivity in changing their strategies (and consequently the online optimization environment they induce). To what extent are the ideas in this result different from prior work in this area - is there something specific to symmetric games (such as faster convergence) that is explored in the paper? More broadly, this raises a concern about the technical depth and significance of the results.

2. I do not understand the comparison to self-play. If I understand correctly, self-play is used to come up with a static strategy (via playing copies of the learner) that is then tested against a particular benchmark strategy for the (n-1) opponents. Importantly, the self-play process does not get to see the opponents' strategy while learning. On the other hand, the no-regret algorithms get to see the opponents' strategy, either fixed or slowly changing, and are allowed to adapt to this strategy. I'm not sure how meaningful this comparison is, since it does not appear to compare the same shape of object.

**Questions:**

My questions are related to the weaknesses highlighted above --

1. To what extent are the ideas in the result about dynamic regret different from prior work in this area. Is there something specific to symmetric games (such as faster convergence) that is explored in the paper? Is there a reason that results about dynamic regret do not directly imply Theorem 5.2.

2. Does it make sense to compare self-play to no-regret (and variants) given that the two frameworks (as set up in the paper) get to see different levels of information? For example, if the experiments adversarially pick the opponent strategy based on the result of self-play, then the comparison is trivial since we would then be comparing playing first (self-play) versus best-responding while playing second (no-regret).

---

> ### Author Response · Authors · 2024-11-21
>
> Thanks a lot for reading our paper and for your insightful comments.
>
> 1.**Technical depth and significance of the results.**
> Please refer to our general response.
>
> 2.**Self-play and clarification of the experiment.**
>
> We would like to provide further clarification of the experiment. Our experiment aims to show some worst cases (adversarially constructed) that meta-algorithms (SP\_scratch, SP\_BC, SP\_BC\_reg) from prior state-of-the-art systems for multiplayer games, which are shown to converge to NEs in the proposed cases, fail to secure an equal share. Aligned with our theoretical analysis, our proposed games assume that all opponents share the same fixed strategy over all rounds without adaptation. This is arguably the most fundamental scenario in which a principled algorithm should work well, given that in reality human opponents can be adaptive or even adversarial.
>
> For self-play-based algorithms, we first run them to convergence and then evaluate the resulting policy against the fixed opponents' strategy. While SP\_scratch cannot access opponents' strategy, we also add SP\_BC and SP\_BC\_reg as baselines for fair comparison, which use the opponents' strategy as initialization or regularization. These design choices originate from modern variants of self-play that achieve state-of-the-art performance on multi-player games. In Section 3 we also highlight that self-play without opponents' strategy can fail easily in simple games such as majority vote.

---

> > ### Author Response · Authors · 2024-11-25
> >
> > Thank you for taking the time to review our paper and provide your thoughtful feedback. We hope our response has addressed your concerns and clarified any uncertainties. As the discussion period is nearing its conclusion, please feel free to share any remaining questions or comments. We would be happy to discuss them further.

---

> > ### Comment · Reviewer_sS4W · 2024-11-26
> >
> > Thanks for your considered response.
> >
> > I appreciate that your hardness results are novel, due to proving it for a subclass of games.
> >
> > I still do not understand the significance of comparing to self-play. Is it clear that the initialization is preserved under the self-play training dynamics is considered? Likewise, does the regularization help to protect against the eventual opponent strategy.

---

> ### Author Response · Authors · 2024-11-26
>
> Thank you for your further thoughtful comments.
>
> 1. We compare our methods to self-play algorithms because our goal is to develop an intelligent agent capable of excelling against other players, particularly humans, in multiplayer games. This high-level objective aligns with prior research on games such as Mahjong, Poker, and Diplomacy, where self-play has been established as the state-of-the-art approach.
>
> 2. In these tasks, understanding opponents' or human meta-strategies (e.g., average human strategies in Mahjong) is crucial for developing more effective AI agents. Prior state-of-the-art studies on games like Mahjong and Diplomacy have proposed leveraging human strategies as initialization or regularization for self-play, demonstrating improved performance compared to starting self-play from scratch. Therefore, prior self-play methods inherently utilize opponents' strategies. Based on this, we believe our comparison is fair, as we did not rely on additional information beyond what is commonly used.
>
> 3. Regarding whether using opponents' strategies as initialization or regularization for self-play guarantees protection against those strategies, we agree with the reviewer that the answer may be "no" for generic multiplayer games. These approaches, as proposed in prior work, are heuristic methods designed to complement self-play and lack theoretical guarantees against opponents' strategies. This precisely highlights the importance of this paper, which develops a principled approach to better leverage the information of opponents' strategy, and provably achieve equal share protecting against opponents' strategy.

---

### Official Review · Reviewer_sdNv · 2024-11-07

**Soundness:** 3
**Presentation:** 3
**Contribution:** 2
**Rating:** 6
**Confidence:** 3

**Summary:**

The paper considers multiplayer symmetric constant-sum games, focusing on achieving equal share - where a player secures an expected payoff of $C/n$ in an n-player game with total payoff $C$. This work is particularly relevant for games like Mahjong, Poker, and Diplomacy, where traditional two-player zero-sum game theory falls short.

The paper makes serveral theoretical contributions. First, they identify two conditions for achieving equal share (over the worst case): all opponents must deploy the same strategy, and they must have limited adaptivity while being modeled by the learning agent.

For fixed opponents, they employ the Hedge algorithm; for slowly adapting opponents, they introduce the SAOL_H algorithm; and for intermediately adapting opponents, they utilize behavior cloning. Importantly, they provide matching lower bounds that demonstrate their algorithms are near optimal.

---------
Post rebuttal: I have read the author's rebuttal. My concerns on the technical novelty remains (this is also observed by other reviewers). I still think the paper studies an interesting topic.

**Strengths:**

The mult-player constant sum game is of great interests to practice, and the strength of this paper paper lies in its conceptual contribution.

**Weaknesses:**

1. The technique is not strong, seems to be a natural modificaiton to the existing (adaptive) online learning algorithm.


Minor issue: Some of the the most up to date (and the state-of-art) literature on hardness of NE and no-regret learning in games:

[1] Inapproximability of Nash equilibrium, Aviad Rubinstein. STOC'15.
(This paper gave the best hardness result for NE)

[2] Fast swap regret minimization and applications to approximate correlated equilibria. Binghui Peng, Aviad Rubinstein. STOC'24

[3] From External to Swap Regret 2.0: An Efficient Reduction for Large Action Spaces. Yual Dagan, Constantinos Daskalakis, Max Fishelson, Noah Golowich. STOC 2024

These two papers gave the best algorithm for learning in normal-form/extensive-form games.

**Questions:**

No.

**Details Of Ethics Concerns:**

No.

---

> ### Author Response · Authors · 2024-11-21
>
> Thanks a lot for reading our paper and for your insightful comments.
>
> 1. **The technique is not strong.**
> Please refer to our general response which discusses the novelty and technical contribution of our work.
>
> 2. **Some of the literature on the hardness of NE and no-regret learning in games.**
> We thank the reviewer for providing these references about state-of-the-art literature on the hardness of NE and no-regret learning. We will include them in the discussion of related work.

---

> > ### Author Response · Authors · 2024-11-25
> >
> > Thank you for taking the time to review our paper and provide your thoughtful feedback. We hope our response has addressed your concerns and clarified any uncertainties. As the discussion period is nearing its conclusion, please feel free to share any remaining questions or comments. We would be happy to discuss them further.

---

### Author Response · Authors · 2024-11-21
**General Response**

We sincerely thank all the reviewers for their thoughtful feedback, valuable insights, and recognition of the need to move beyond equilibrium-based approaches. In response to the concerns regarding technical contributions raised by multiple reviewers, we would like to emphasize that identifying the appropriate goal (i.e., achieving equal share) and proposing corresponding solution concepts—specifically, understanding under which scenarios the proposed goal can be achieved and determining the correct solutions to consider—is as crucial, if not more so, than developing advanced techniques for well-defined, existing problems. By focusing on establishing a solid foundational framework, we hope our paper serves as an initial step toward advancing principled methodologies and algorithms for multiplayer games.

In the following, we provide further details regarding the novelty and technical contributions of our work.

**Novelty**

In multiplayer games, much of the existing research has concentrated on achieving equilibrium-based outcomes. However, relatively few studies have focused on introducing new solution concepts and principled algorithms. Equilibrium-based approaches, though widely studied, have long been known to have limitations in multiplayer contexts, often making them unsuitable as comprehensive solution concepts. Our work shifts the focus beyond equilibrium as the sole objective, instead introducing an alternative approach centered on symmetry—specifically, achieving an equal share of rewards. From this perspective, we propose new solution concepts and provide principled algorithms explicitly designed to achieve equitable outcomes rather than strictly equilibrium-based solutions. We hope this work will inspire further research in this direction, contributing to the future development of superhuman AI capable of navigating and making decisions in generic complex multiplayer environments.

**Technical Contribution on Efficient Algorithms**

In addition to our contribution to identifying the correct solution concept, this paper also provides a comprehensive set of algorithmic results for achieving equal share in various settings: (1) fixed opponents; (2) slowly changing opponents; (3) opponents that adapt at intermediate rates; (4) matching lower bounds.

- For (1) and (2), we leverage established techniques from the no-regret and no-dynamic-regret learning literature, achieving equal share with provable guarantees. Our contribution here mostly lies in identifying and bringing the right tools and adapting them to solve the right setup of the multiplayer games, where, to our best knowledge, prior SOTA AI systems on multiplayer games (for Poker, etc) have not utilized techniques such as no-dynamic-regret.
- For (3), we believe our discovery here is completely new, that simple algorithms like behavior cloning can sometimes even outperform sophisticated no-dynamic regret algorithms. Such a result is only possible by leveraging the symmetric structure of the multiplayer game beyond treating it as one versus an adversarial group of opponents.
- For (4), while similar lower bounds have been shown in more general games, they do not apply to our settings as the hard instances constructed in prior lower bounds are not symmetric games. In this paper, we carefully construct new hard instances showing the algorithmic results we proved in (2) and (3) are near-optimal.

---

### Meta-Review · Area_Chair_Qu7z · 2024-12-20

**Metareview:**

The paper proposes a framework for multiplayer symmetric constant-sum games by introducing the concept of equal share, where a player secures an expected payoff of C/n in an n-player game with total payoff C. The reviewers generally found the problem well-motivated and welcomed work in this space. However, there was a consistent sentiment that the technical contributions and experimental comparisons lacked sufficient novelty and depth to merit acceptance at this stage.

All reviewers appreciated the effort to move beyond equilibrium-based methods, which have limitations in multiplayer games. Reviewer 5Z1z noted that the paper “contributes to this line of work by proposing a natural objective,” while reviewer sdNv described the conceptual contribution as a “strength of the paper.” However, multiple reviewers, including 5Z1z and sS4W, expressed concerns about the novelty of the technical results. Reviewer sS4W questioned, “To what extent are the ideas in the result about dynamic regret different from prior work in this area?” Similarly, reviewer sdNv remarked that the techniques seem “a natural modification to the existing (adaptive) online learning algorithms.” The authors included in their general answer, a rebuttal of the novelty concerns, but the reviewers did not change their mind.

The experimental results also drew mixed reactions. Reviewer sS4W appreciated the worst-case analysis highlighting the limitations of self-play but critiqued the fairness of the comparison, stating, “I’m not sure how meaningful this comparison is, since it does not appear to compare the same shape of object.” Reviewer Yqqg raised concerns about the broader applicability of the results, remarking that “these results may not help solve general multiplayer games.” The authors provided a response, but the reviewer maintained a negative opinion.

Ultimately, while reviewers recognized the potential of the proposed framework, the paper’s current scope and contributions were deemed insufficient for acceptance. Reviewer 5Z1z summarized this sentiment by stating, “the technical contributions of this paper do not go far enough.”

**Additional Comments On Reviewer Discussion:**

The discussion focused on the novelty of the proposed framework, the technical contributions, and the fairness of experimental comparisons.

The reviewers debated the fairness of comparing the proposed methods to self-play. The authors argued that their setup was consistent with prior work, stating, “We believe our comparison is fair, as we did not rely on additional information beyond what is commonly used.” However, reviewer sS4W maintained skepticism ("I still do not understand the significance of comparing to self-play").

Finally, Reviewer Yqqg questioned the broader applicability of the framework, noting that “the large games mentioned in this paper are not symmetric.” The authors addressed this by explaining a transformation process to symmetrize games through randomized roles, which they argued ensures fairness in the analysis.

Despite the authors’ clarifications, the reviewers remained unconvinced that the contributions sufficiently advanced the state of the art, as summarized above.

---

### Decision · Program_Chairs · 2025-01-22

Reject